# Digital wearable insole-based identification of knee arthropathies and gait signatures using machine learning

Matthew F Wipperman[1,2]*[†], Allen Z Lin[3†], Kaitlyn M Gayvert[3†],
Benjamin Lahner[1,2,3], Selin Somersan-Karakaya[2], Xuefang Wu[4], Joseph Im[4],
Minji Lee[3], Bharatkumar Koyani[4], Ian Setliff[3], Malika Thakur[4], Daoyu Duan[1,2],
Aurora Breazna[5], Fang Wang[1,2], Wei Keat Lim[3], Gabor Halasz[3], Jacek Urbanek[5],
Yamini Patel[6], Gurinder S Atwal[3], Jennifer D Hamilton[1,2], Samuel Stuart[4],
Oren Levy[2], Andreja Avbersek[2], Rinol Alaj[4]*, Sara C Hamon[1,2]*, Olivier Harari[2]*

[1]Precision Medicine, Regeneron Pharmaceuticals Inc, Tarrytown, United States;
[2]Early Clinical Development & Experimental Sciences, Regeneron Pharmaceuticals
Inc, Tarrytown, United States; [3]Molecular Profiling & Data Science, Regeneron
Pharmaceuticals Inc, Tarrytown, United States; [4]Clinical Outcomes Assessment
and Patient Innovation, Global Clinical Trial Services, Regeneron Pharmaceuticals
Inc, Tarrytown, United States; [5]Biostatistics and Data Management, Regeneron
Pharmaceuticals Inc, Tarrytown, United States; [6]General Medicine, Regeneron
Pharmaceuticals Inc, Tarrytown, United States

*For correspondence:
matthew.wipperman@regeneron.
com (MFW);
rinol.alaj@regeneron.com (RA);
sara.hamon@regeneron.com
(SCH);
olivier.harari@regeneron.com
(OH)

[†]These authors contributed
equally to this work

Competing interest: See page
20

Reviewing Editor: Hristo
Dimitrov, University of
Cambridge, United Kingdom

**Abstract** Gait is impaired in musculoskeletal conditions, such as knee arthropathy. Gait analysis is used in clinical practice to inform diagnosis and monitor disease progression or intervention response. However, clinical gait analysis relies on subjective visual observation of walking as objective gait analysis has not been possible within clinical settings due to the expensive equipment, large-scale facilities, and highly trained staff required. Relatively low-cost wearable digital insoles may offer a solution to these challenges. In this work, we demonstrate how a digital insole measuring osteoarthritis-specific gait signatures yields similar results to the clinical gait-lab standard. To achieve this, we constructed a machine learning model, trained on force plate data collected in participants with knee arthropathy and controls. This model was highly predictive of force plate data from a validation set (area under the receiver operating characteristics curve [auROC] = 0.86; area under the precision-recall curve [auPR] = 0.90) and of a separate, independent digital insole dataset containing control and knee osteoarthritis subjects (auROC = 0.83; auPR = 0.86). After showing that digital insole-derived gait characteristics are comparable to traditional gait measurements, we next showed that a single stride of raw sensor time-series data could be accurately assigned to each subject, highlighting that individuals using digital insoles can be identified by their gait characteristics. This work provides a framework for a promising alternative to traditional clinical gait analysis methods, adds to the growing body of knowledge regarding wearable technology analytical pipelines, and supports clinical development of at-home gait assessments, with the potential to improve the ease, frequency, and depth of patient monitoring.

## Editor's evaluation

This study presents a valuable dataset and tool that can aid in arthropathies' assessment, potentially enabling such evaluation to be done outside the lab. There is convincing evidence supporting the comparison between the force plate and insole data but the evidence for distinguishing disease

signatures is inconclusive and would need further development. This work will be of interest to physical therapists, clinicians, and researchers in the field of lower limb joint diseases.

## Introduction

Gait assessment plays several roles in clinical practice and research for many musculoskeletal and orthopedic diseases, such as diagnosis, guiding treatment selection, and measuring intervention response. (*Lord et al., 1998*). Knee osteoarthritis (OA) contributes to altered gait as individuals try to avoid knee pain and cartilage contact stress (i.e., weight shift or joint loading onto the non-affected limb) (*Iijima et al., 2019*; *Kaufman et al., 2001*). Those with knee OA commonly have increased lateral trunk lean toward the ipsilateral limb, along with non-significantly increased trunk/pelvic flexion and resultant significant alterations in external hip adduction moments (*Iijima et al., 2019*). Clinically, gait is examined in knee OA through clinical visual observation of walking, which is subjective and dependent on expertise. Beyond visual observation, gait has been traditionally objectively assessed in specialized gait laboratories with expensive equipment, such as force platforms, with or without a motion-tracking system. However, gait laboratories are generally not available or feasible within clinical settings due to cost, need for highly trained staff to operate equipment, and the size of equipment. Wearable sensors offer an alternative approach to assessment of gait in those with knee OA (*Mills et al., 2013*) as they can be deployed within any environment or setting (*Stern et al., 2022*), are relatively low-cost, and can provide outcomes automatically without the need for highly experienced or trained staff.

Vertical ground reaction force (vGRF) is a gait characteristic that is impaired within knee OA as it relates to the bilateral weightbearing capabilities of the patient (i.e., greater peak vGRF) (*Creaby et al., 2013*; *Davis et al., 2019*; *Trentadue et al., 2021*). Traditionally, vGRF is objectively examined using force plates, which can provide three components of force (vertical, anterior-posterior, and medio-lateral) (*Creaby et al., 2013*; *Davis et al., 2019*; *Trentadue et al., 2021*). Force plate signals can provide information on gait characteristics, postural stability, as well as direction, strength, and duration of stance phase (*Arslan et al., 2019*; *Whittle, 2007*). However, force plates only capture intermittent data on vGRF (single or several steps) within controlled experiments that do not represent real-world overground walking. Technological development has led to digital insoles (wearables) that can capture vGRF and other gait characteristics that are relevant to knee OA assessment, which could be used within free-living environments (*Stern et al., 2022*). The data generated from digital insoles may allow for phenotyping information to characterize patients with knee OA. However, clinical application of wearable digital insoles for gait analysis in knee OA is limited by a paucity of analytical validation data (including face, criterion, and construct validity), such as comparison of gait outcome measures in those with knee OA to 'gold-standard' laboratory references (e.g., force plate outcomes) (*Goldsack et al., 2020*; *Rochester et al., 2020*).

Gait quantitation generates dense raw sensor time-series data with nonlinear relationships that make analysis and interpretation challenging. As a class, digital gait data analysis pipelines suffer from a lack of well-established analytical methods compared to other biomarker data types (*Wipperman et al., 2022*; *Crouthamel et al., 2021*; *Horst et al., 2019*). Recently, there is interest in developing more streamlined ('light-weight') gait algorithms and data processing pipelines using machine learning (ML) to automate and process large volumes of novel digital data obtained from wearable devices (*Celik et al., 2022*; *Godfrey et al., 2018*). An ML framework may be used as a tool to evaluate the digital gait outcome quality and consistency, as well as how well these data can be used as potential clinical trial endpoints. Selection of the appropriate modeling modalities for a particular clinical question is challenging (*Horst et al., 2019*; *Slijepcevic et al., 2022*). The selection of classical ML versus deep learning methods may be influenced by the structure and size of the data. For example, deep learning models are better suited to handle high-throughput, multimodal data streams, such as raw sensor time series (*Alias, 2018*; *Briouza et al., 2021*), but typically require larger datasets than classical statistical or ML methods (in terms of both features and sample size). Finally, clinical wearable data may be collected over several seconds to minutes, and longer in passive monitoring settings; thus, appropriate understanding of large dataset processing is important. Collectively, both data type, size, and model selection are key components of a comprehensive wearable sensor data analysis

**eLife digest** The way we walk – our 'gait' – is a key indicator of health. Gait irregularities like limping, shuffling or a slow pace can be signs of muscle or joint problems. Assessing a patient's gait is therefore an important element in diagnosing these conditions, and in evaluating whether treatments are working.

Gait is often assessed via a simple visual inspection, with patients being asked to walk back and forth in a doctor's office. While quick and easy, this approach is highly subjective and therefore imprecise. 'Objective gait analysis' is a more accurate alternative, but it relies on tests being conducted in specialised laboratories with large-scale, expensive equipment operated by highly trained staff. Unfortunately, this means that gait laboratories are not accessible for everyday clinical use.

In response, Wipperman et al. aimed to develop a low-cost alternative to the complex equipment used in gait laboratories. To do this, they harnessed wearable sensor technologies – devices that can directly measure physiological data while embedded in clothing or attached to the user. Wearable sensors have the advantage of being cheap, easy to use, and able to provide clinically useful information without specially trained staff.

Wipperman et al. analysed data from classic gait laboratory devices, as well as 'digital insoles' equipped with sensors that captured foot movements and pressure as participants walked. The analysis first 'trained' on data from gait laboratories (called force plates) and then applied the method to gait measurements obtained from digital insoles worn by either healthy participants or patients with knee problems.

Analysis of the pressure data from the insoles confirmed that they could accurately predict which measurements were from healthy individuals, and which were from patients. The gait characteristics detected by the insoles were also comparable to lab-based measurements – in other words, the insoles provided similar type and quality of data as a gait laboratory. Further analysis revealed that information from just a single step could reveal additional information about the subject's walking.

These results support the use of wearable devices as a simple and relatively inexpensive way to measure gait in everyday clinical practice, without the need for specialised laboratories and visits to the doctor's office. Although the digital insoles will require further analytical and clinical study before they can be widely used, Wipperman et al. hope they will eventually make monitoring muscle and joint conditions easier and more affordable.

pipeline and should be considered when evaluating clinical research pipelines with large heterogeneous datasets.

This study aimed to assess the face validity of measuring vGRF using digital insoles compared to the standard laboratory reference of force plates in control adults and those with knee arthropathy. We also aimed to clinically validate the measurement of gait with digital insoles in those with knee arthropathy through (1) development of a novel ML framework (based on 'gold-standard' force plate data) for application within digital insoles to detect knee arthropathy status and (2) processed digital insole-derived gait outcomes and raw sensor signals to identify disease-specific gait patterns in knee arthroscopy. Clinical validity is the ability to provide clinically meaningful outcome measures, including the detection of disease-specific patterns relative to controls, and the identification of subject-specific gait patterns.

## Results

### Platform-agnostic visualization of knee arthropathy signatures from vGRF data

vGRF is the major and clinically relevant component of the ground reaction forces generated from walking and can be measured using force plates or digital insoles. We obtained vGRF data from the three studies: the GaitRec Force Plate study of subjects with knee injuries (N = 625) and controls (N = 211) and two digital insole studies in controls (N = 22) and subjects with knee OA (N = 40), respectively (*Figure 1*, *Figure 2—figure supplement 1*). We plotted raw and normalized vGRF values recorded by force plate and digital insole devices and observed by visual inspection that the means of vGRF

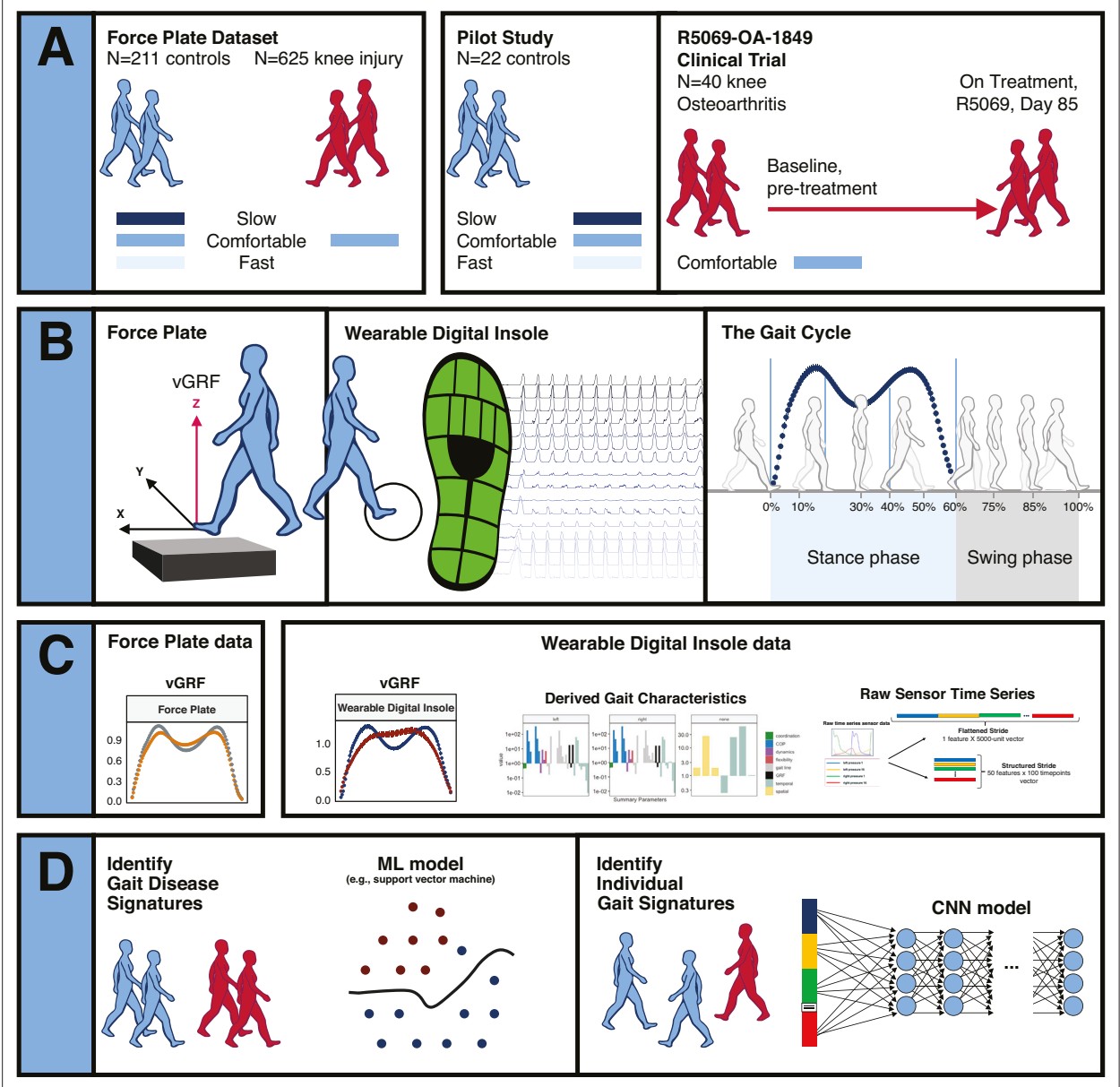

**Figure 1.** Overview of data sources and study participants, device types, data types, and clinical research questions. (**A**) Three datasets were used for analyses. The GaitRec force plate dataset (force plate data) contains N = 211 controls, who walked at three different walking speeds (slow, comfortable, and fast), and N = 625 knee injury subjects, who walked at a comfortable walking speed (**Horsak et al., 2020**). The second dataset is from a digital insole pilot study, where N = 22 controls walked at three different walking speeds (slow, comfortable, and fast). The third dataset is from a digital insole sub-study from a longitudinal clinical trial in knee osteoarthritis (OA), where N = 40 knee OA subjects performed a 3 min walk test (3MWT) at a comfortable walking speed at baseline (pretreatment) and at day 85 (on treatment). (**B**) Both force plates and digital insoles produce data collected during stance and swing phases of a person's gait cycle. (**C**) Types of data produced by these devices include vertical ground reaction force (vGRF), derived gait characteristics, and raw sensor time series. (**D**) Clinical research questions addressed in this work include the derivation of gait disease signatures of knee OA and investigation of the individuality and consistency of gait patterns. Two analytical methods were used to evaluate these data. Support vector machine (SVM) models were used to analyze vGRF, derived gait characteristics, and raw sensor time-series flattened stride data. A one-dimensional convolutional neural network (CNN) was used to analyze structured stride raw sensor time-series data.

curves within each disease category are similar across platforms (**Figure 2A**), which highlights face validity of the digital insoles (i.e., the insole data appeared similar to the force plate data). Furthermore, we observed distinct patterns between control subjects and those with knee arthropathy when comparing the normalized vGRF values averaged by group (**Figure 2A**) or per individual (**Figure 2B**). Specifically, individuals with knee arthropathy from all evaluated datasets displayed a qualitatively

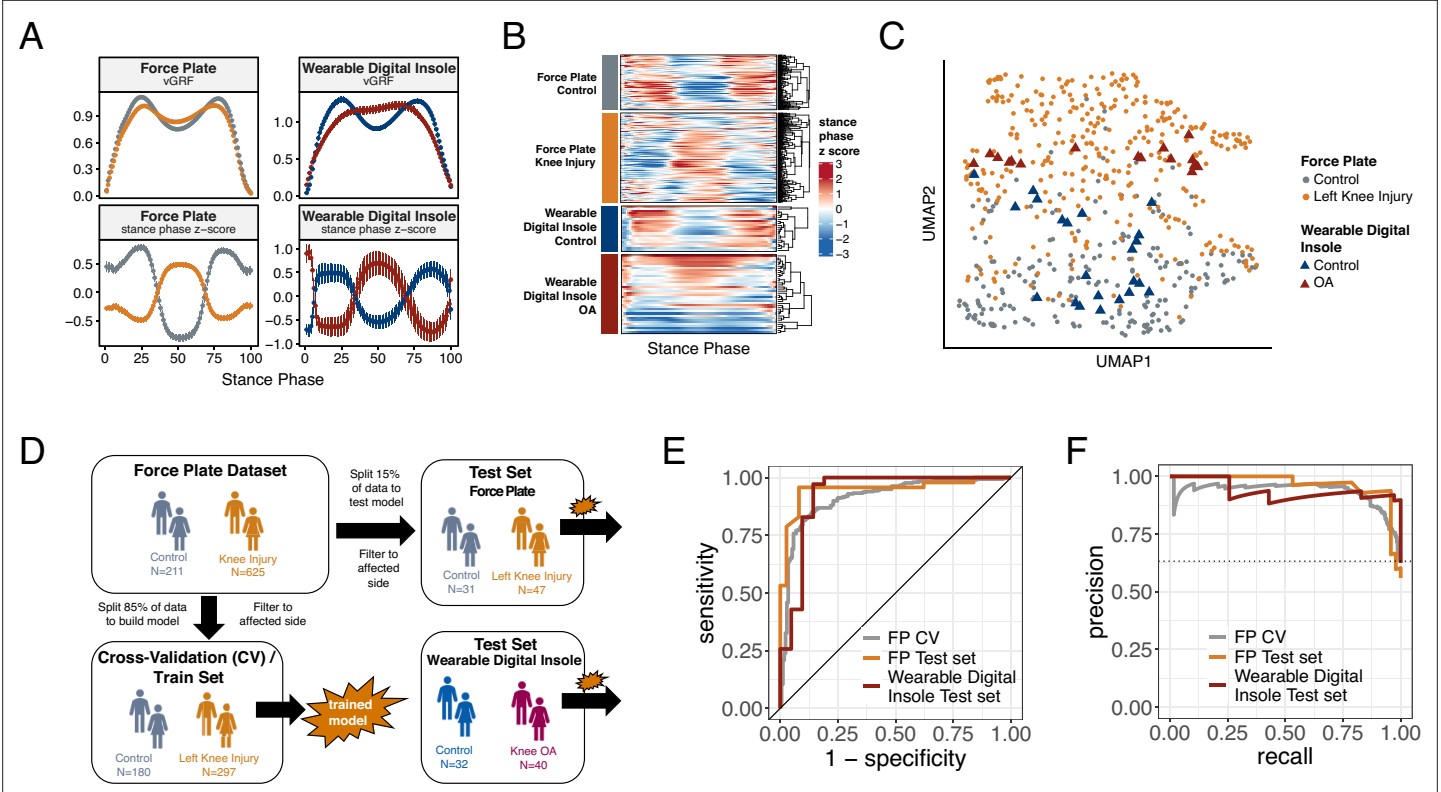

**Figure 2.** Machine learning (ML) model trained on knee injury subjects walking on force plates accurately classifies osteoarthritis (OA) patients wearing digital insoles. (**A**) Vertical ground reaction force (vGRF) curves derived from force plate and digital insole data for controls, and knee injury and knee OA patients, respectively. Left foot data are shown as mean of values (top panels) and mean of normalized z-scores (bottom panels) at each percent stance phase within each device and health status. Groups are color-coded as in (**B**) and (**C**). (**B**) vGRF curves for an individual's left foot shown as heatmap rows, after data was z-transformed at each percent stance phase (as in **A**). Rows are hierarchically clustered within each group of subjects. (**C**) Uniform Manifold Approximation and Projection (UMAP) dimensionality reduction of the z-transformed left foot vGRF data. Each point represents a subject, and points are colored by phenotype, and shaped by device. (**D**) Schematic of machine learning model building of training/validation and testing sets. Two support vector machine (SVM) models were created, one for left knee injury (depicted) and one for right knee injury. The full force plate vGRF dataset with both controls (comfortable walking speed) and left or right knee injury subjects (comfortable walking speed, excluding subjects with knee injury on both joints) were split 85% into training/validation datasets, and 15% into a hold-out testing set. One model predicts control versus knee injury subjects using left foot data (of left knee injury subjects and all controls), and the other predicts using right foot data (of right knee injury subjects and all controls). These models were then applied on a separate, independent testing set of digital insoles vGRF data with N = 22 control subjects and N = 38 patients with knee OA. (**E**) Receiver operating characteristic curve for SVM classification of force plate (85%) cross-validation (CV, training/validation) set, force plate (15%) hold-out test set, and the digital insole test set. (**F**) Precision-recall curve for SVM classification of the same groups in (**E**).

The online version of this article includes the following figure supplement(s) for figure 2:

**Figure supplement 1.** Heatmap representation of vertical ground reaction force (vGRF) data from GaitRec dataset for all joints with injuries and controls (*Horsak et al., 2020*).

**Figure supplement 2.** Variance explained in vertical ground reaction force (vGRF) with clinical and demographic characteristics of the participants.

**Figure supplement 3.** Model results from the right foot only data.

**Figure supplement 4.** Comparison between logistic regression, support vector machine, and XGBoost models.

different gait signature than controls, with a 'flatter' vGRF curve shape during the middle of stance phase (we use the term 'arthropathy' to encompass both knee injury and OA).

Using a dimension reduction approach Uniform Manifold Approximation and Projection (UMAP) with normalized vGRF data from each subject in two dimensions, subjects at a population level separated out by arthropathy status (knee arthropathies vs controls) rather than by measurement platform (*Figure 2C*). Thus, despite performing analysis on data collected from different devices at different sites, we could discern disease-relevant patterns in the vGRF data and show that the digital insole data recapitulated the force plate data.

**Table 1.** Force plate vertical ground reaction force (vGRF) control versus knee arthropathies support vector machine (SVM) classification model evaluation statistics (left foot/right foot).

An SVM model was trained on 85% of the force plate dataset vGRF data to predict control or knee arthropathies (knee injury or knee osteoarthritis) classes, with left foot vGRF data used to predict left knee arthropathies and right foot vGRF data used to predict right knee arthropathies. The model was evaluated using fivefold cross-validation, a hold-out force plate test set, and a digital insole test set. Area under the receiver operating characteristics curve (auROC) and area under the precision-recall curve (auPR) statistics are reported for the three models. F1 scores for each class for each model are also reported.

| | auROC actual/mean (SD) | | auPR actual/mean (SD) | | F1 score | | | |
| | | | | | Control | | Arthropathy | |
| Per class binary classification metrics | Left | Right | Left | Right | Left | Right | Left | Right |
| --- | --- | --- | --- | --- | --- | --- | --- | --- |
| Force plate fivefold cross validation | 0.917 (0.034) | 0.937 (0.023) | 0.944 (0.029) | 0.960 (0.015) | 0.78 | 0.84 | 0.87 | 0.88 |
| Force plate test set | 0.949 | 0.935 | 0.955 | 0.950 | 0.78 | 0.76 | 0.87 | 0.87 |
| Digital insole test set | 0.928 | 0.925 | 0.937 | 0.938 | 0.89 | 0.83 | 0.95 | 0.90 |

SD, standard deviation.

To investigate how the variation in the vGRF data may be partially explained by the clinical and demographic characteristics of the participants, we fit a series of linear models to each point along the vGRF curve, with arthropathy state (knee arthropathy or control), age, sex (male or female), and body weight as covariates in the model (*Figure 2—figure supplement 2*). For each linear model, representing the sum of squares for each category compared to the total sum of squares as a percentage of variation explained by that component, we observed that the arthropathy state is the major contributor to the vGRF signal for most of the vGRF curve, with age, sex, and body weight explaining a smaller proportion of the variance. We conclude that the dominant factor likely contributing to variation among participants to these signals is arthropathy state.

## ML models trained on vertical ground reaction force plate data to classify control versus knee injury across different platforms

We next looked to quantify how well force plate data can identify disease using gait signatures and to understand if a wearable insole could detect similar characterizations. To differentiate controls versus knee arthropathies using vGRF data, we divided the complete vGRF force plate dataset into a training/validation set (85%) and a test set (15%) and trained a support vector machine (SVM) model to predict these classes (*Figure 2D*). The model indicated strong predictive power to classify control versus knee injury subjects using force plate data when evaluated using fivefold cross-validation of the training/validation dataset, a standard method of assessing model performance (left knee: area under the receiver operating characteristics curve [auROC] = 0.92 [SD = 0.03], area under the precision-recall curve [auPR] = 0.94 [SD = 0.03]; right knee: auROC = 0.94 [SD = 0.02], auPR = 0.96 [SD = 0.01]). The predictive power was also strong when assessed on the hold-out test dataset, which was not used for training of the SVM model (left knee: auROC = 0.95, auPR = 0.95; right knee: auROC = 0.93, auPR = 0.95) (*Table 1* and *Figure 2—figure supplement 3*).

To further assess generalizability of the model and understand if a digital insole could measure vGRF similarly to a force plate, we applied the model trained using vGRF force plate data to the separate, independent dataset derived from digital insole studies on individuals with knee OA and controls. An important assumption of this analysis was that knee OA and knee injury are represented similarly by vGRF curves as was shown earlier (*Figure 1A–C*). We found that for the digital insole an SVM model trained only on force plate data performed as well on the digital insole data (left knee: auROC = 0.93, auPR = 0.94; right knee: auROC = 0.93, auPR = 0.94) as it did on the hold-out force plate test data (*Figure 2E and E*; *Table 1*, and *Figure 2—figure supplement 3*).

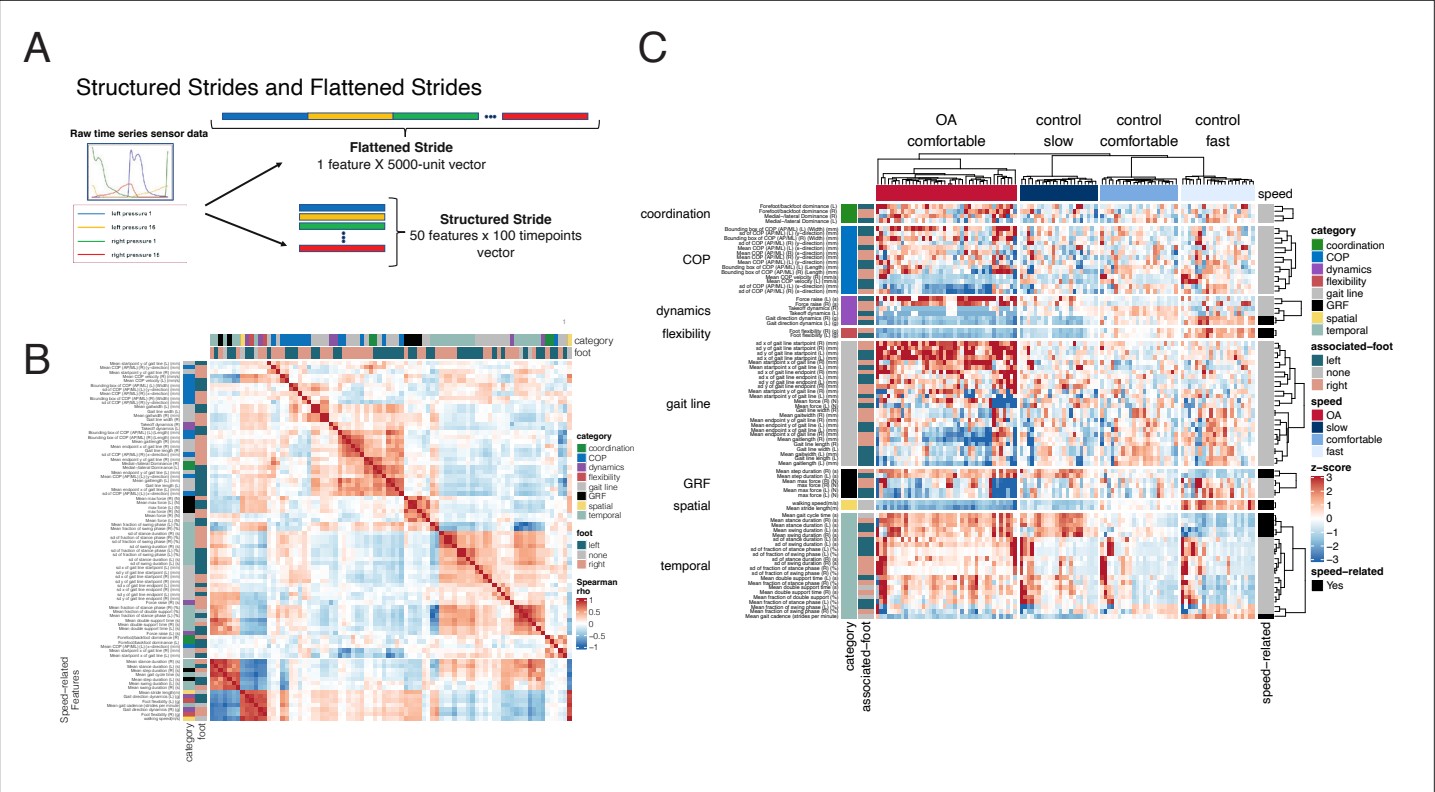

**Figure 3.** Derived gait characteristics from a digital insole measured across all subjects from the pilot study and knee osteoarthritis patients in the clinical trial. (**A**) Schematic of raw sensor time-series data from a digital insole. Data can be processed from the device in three ways: (1) vertical ground reaction forces (***Figure 1***); (2) derived gait characteristics on force, spatio-temporal, and center of pressure aspects; and (3) raw sensor time-series data from the 50 sensors embedded across both insoles. Each segmented stride of raw sensor time-series data can be analyzed as is (structured strides) or collapsed (flattened strides). (**B**) The derived gait characteristics (parameters) of the digital insole from all individuals in the pilot study were correlated against each other at the comfortable walking speed. Spearman correlation coefficients were computed and shown in a correlation matrix ranging from –1 (perfect anti-correlation) to +1 (perfectly correlation). Each parameter has a Spearman correlation coefficient of +1 with itself (red diagonal). The parameter, the foot from which it was generated, and its category are labeled on the left of the correlation matrix. (**C**) Heatmap representation of the average of each of the 82 digital insole parameters (rows) across all walks for each patient (columns) from the pilot study. Parameter values are shown as normalized z-scores (bounded within ± 3), calculated across all participants, and walking speeds. The heatmap is split by the three walking speeds (slow, normal, fast), and columns are clustered within each walking speed using hierarchical clustering with Euclidean distances. The 14 parameters strongly correlated with walking speed are indicated on the right of the heatmap.

The online version of this article includes the following figure supplement(s) for figure 3:

**Figure supplement 1.** Derived gait characteristics that are most discriminative of knee osteoarthritis (OA) versus controls include features shown in ***Supplementary file 1***.

## Performance of derived gait characteristics of the digital insole versus vGRF and raw sensor time series for disease classification

A potential benefit of a digital insole relative to a force plate is that many more variables can be derived, thus enabling a more comprehensive assessment of gait. For instance, in addition to the vGRF curves, derived gait characteristics and raw sensor time-series data can be obtained using the 50 sensors across both insoles (***Figure 3A***).

Time-series data measures different aspects of a stride, including force, angular velocity, and orientation of the foot relative to gravity. The raw sensor time-series data can be evaluated as either structured strides, in which each stride is represented such that each sensor's values are measured over time, or converted into flattened strides, in which all timepoints and sensors are concatenated into one representation (***Figure 3A***) providing a linear representation of a person's stride.

To investigate how derived gait characteristics relate to each other, we clustered the values and their correlations across different walking speeds and disease status (OA vs control). Correlations within and between categories of parameters revealed that similar groups of parameters clustered

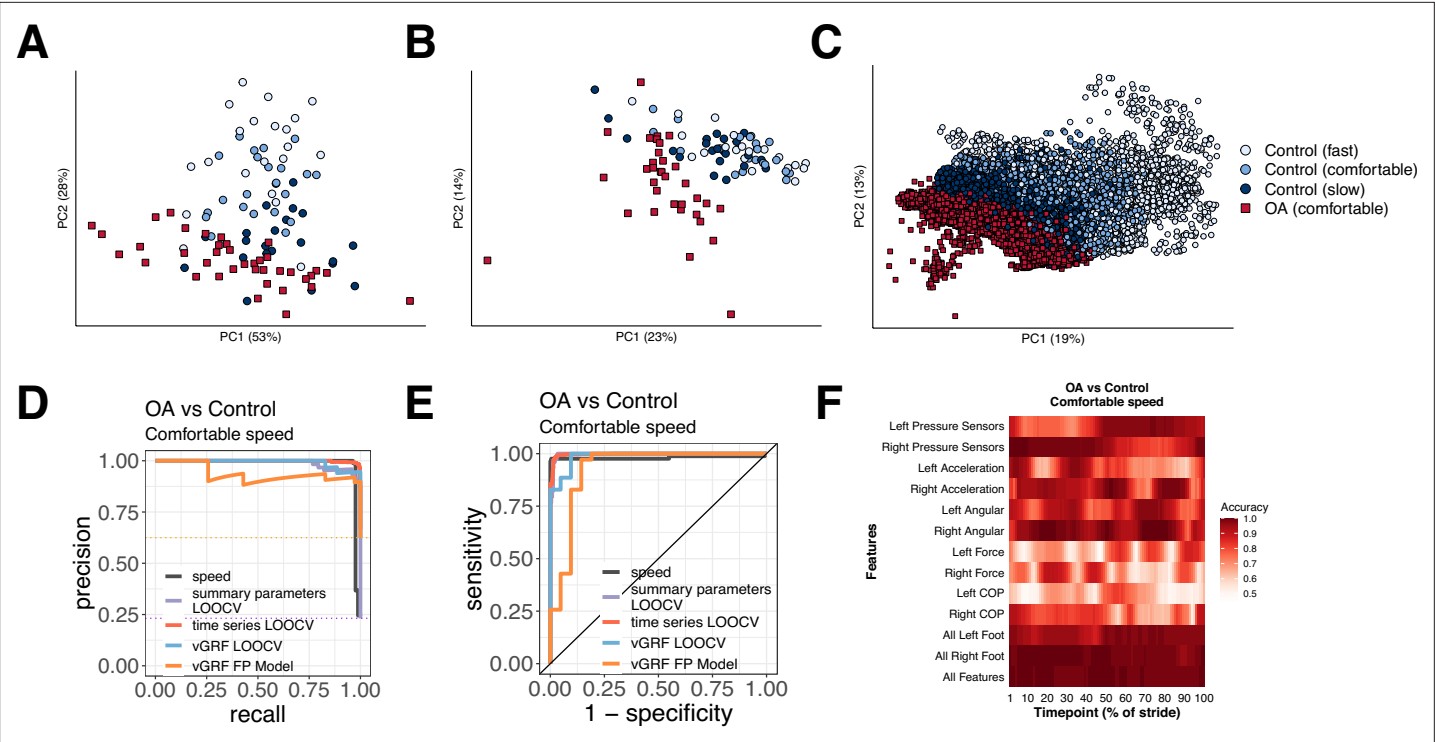

**Figure 4.** Different methods to analyze control subject versus knee osteoarthritis (OA) patient data from a digital insole enable refined classification of disease signatures. (**A**) Principal component analysis (PCA) dimensionality reduction of vertical ground reaction force (vGRF) data from all walks of pilot study subjects and baseline walks of knee OA clinical trial patients. Each dot represents data from a single subject at a given walking speed. (**B**) PCA dimensionality reduction of derived gait characteristic data from the digital insole, without the 14 speed-correlated derived gait characteristics. (**C**) PCA dimensionality reduction of raw sensor time series of each stride from all walks. Each dot represents data from a single stride and repeat strides from the same participant are shown. (**D**) Receiver operating characteristic curves for knee OA versus control (both at comfortable walking speed) prediction using only walking speed (speed), derived gait characteristics (excluding 14 speed-correlated features), raw sensor time series, and vGRF. Classification metrics were derived using leave-one-out cross-validation (LOOCV). The single derived gait characteristic speed separates out digital insole knee OA patients versus control subjects. (**E**) Precision-recall curves of the same comparisons in (**D**). (**F**) Classification accuracy using raw sensor time-series data from control subjects versus knee OA patients using subsets or all 50 sensors at each timepoint of the stride (0–100% of the stride). Timepoints start with the stance phase of the right foot and swing phase of the left foot, and end with the swing phase of the right foot and the stance phase of the left foot. Classification accuracy of 1.0 indicates perfect knee OA versus control classification using data from that timepoint.

The online version of this article includes the following figure supplement(s) for figure 4:

**Figure supplement 1.** Evaluation of all speed-independent characteristics for OA vs control classification.

together (*Figure 3B*), including 14 derived gait characteristics strongly correlating with walking speed (|Spearman rho| > 0.7). We note that many other gait characteristics will be influenced by walking speed, and as such, we call out these as representing the subset most influenced by speed, defined by this threshold (|Spearman rho| > 0.7). Using a heatmap, where derived gait characteristics were normalized across both control and knee OA populations, we observed distinct patterns between derived gait characteristics at different walking speeds and disease status (*Figure 3C*). To further explore the relationship with walking speed, we performed a principal component analysis (PCA) dimensionality reduction on each data type. This analysis demonstrated that knee OA arthropathy state can be observed on a continuum related to walking speed. Compared to control subjects, participants with knee OA are walking more slowly as apparent across all data types, including vGRF (*Figure 4A*), derived gait characteristics (*Figure 4B*), and raw sensor time-series data (*Figure 4C*).

We revisited the question of whether gait data can be used to identify arthropathy status relative to control subjects for each of the three types of data collected by the digital insole: vGRF, summary parameters, and time-series data. Note that this analysis aims to classify whether a subject has knee arthropathy, rather than to determine the severity of arthropathy, which this study was not designed to assess. SVM models were trained on vGRF and assessed using both repeated five-fold cross-validation (r5FCV) and leave-one-out cross-validation (LOOCV), where models were evaluated

by iteratively leaving one subject out, building a model, and evaluating where that subject would be classified compared to the true result (see 'Methods'). This was also performed for derived gait characteristics, and raw sensor time-series (flattened strides) data independently (*Figure 4D and E*). Additionally, we used walking speed as a single variable predictor of knee OA (both control subjects and OA patients had been asked to walk at a self-paced comfortable walking speed). We found that walking speed alone was able to discriminate between knee OA subjects and controls (auROC = 0.981, auPR = 0.983). vGRF with the digital insole also demonstrated high predictive power (LOOCV: auROC = 0.984, auPR = 0.990; r5FCV: auROC = 0.988, auPR = 0.992).

For derived gait characteristics, we wanted to understand whether aspects of gait other than walking speed could be used to correctly classify whether a subject had knee OA relative to a control as this would suggest potential for broader applicability if disease-specific features in addition to changes in speed could be detected with digital insoles. Using derived gait characteristics that excluded the 14 characteristics strongly correlated to walking speed, we found even better classification accuracy (LOOCV: auROC = 0.997, auPR = 0.988; r5FCV: auROC = 0.996, auPR = 0.986). The most important discriminating parameters included takeoff dynamics, max force (N), mean COP velocity (mm/s), and gait line-associated parameters (sd x and sd y of gait line start point [mm]) (*Supplementary file 1*, and *Figure 3—figure supplement 1*). Flattened strides from raw sensor data, which is also independent of walking speed as each stride was interpolated to a consistent 100 timepoints, also were predictive (LOOCV: auROC = 0.997, auPR = 0.998) (*Figure 4—figure supplement 1*).

Finally, we sought to evaluate the contribution of each sensor at each timepoint along individual segmented strides to the disease classification accuracy. We trained additional SVM models on subsets of sensor type at each timepoint (*Figure 4F*) and found that classification accuracy for control versus knee OA depends on the type of sensor, the timepoint along a stride, and the foot (left vs right). Measurements of pressure and force from a foot had greater mean classification accuracy during the stance phase of that foot.

## Deriving individual gait signatures using convolutional neural net latent representations of raw sensor time-series data or derived gait characteristics

To determine the extent to which walking patterns are specific to an individual, subjects were split 50:50 into training and testing sets and stratified by disease status (*Figure 5A*). We then trained a one-dimensional convolutional neural net (CNN) on structured strides of training set individuals and subsequently applied the CNN model on structured strides of testing set individuals. For each stride, we extracted the 60 features in the last connected (penultimate) layer of the CNN (*Figure 5B*). This layer directly precedes the final output of the CNN model predicting the individual from which the stride came, and thus these 60 features constitute a gait 'fingerprint' learned by the CNN model. These features were learned by the CNN to distinguish individuals, and thus these patterns (captured in the latent features) can subsequently be used for classification of new subjects, previously unseen by the model.

To visualize the latent representation of strides from the CNN, a UMAP clustering of these latent features from each stride indicated that this representation captured the individual identity of participants (*Figure 5C*) as strides from the same individuals clustered together in both the training and testing sets.

We sought to quantify the individuality of each of the three representations of gait: derived gait characteristics from each walk, raw (flattened) time series of each stride, and CNN latent features of each stride. To do so, we calculated the distance between all pairs of test set walks/strides in such representation against each other (see 'Methods'). These distances are displayed in both heatmaps (*Figure 5D*, top) and boxplots (*Figure 5D*, bottom). An ideal representation to quantitate individuality would have low distances between walks/strides from the same person and high distances for between walks/strides from different people (*Figure 5—figure supplement 1*).

Subjects within their class (control or knee OA) displayed similar gaits; therefore, we separated out comparisons of subjects to their same class versus comparisons to the other class (with other control, with other OA). All three representations had significant differences in distances when comparing strides from the same subjects versus strides from different subjects (p<0.001, *t*-test). The CNN latent representation was best at minimizing distances of strides from the same subjects while maximizing

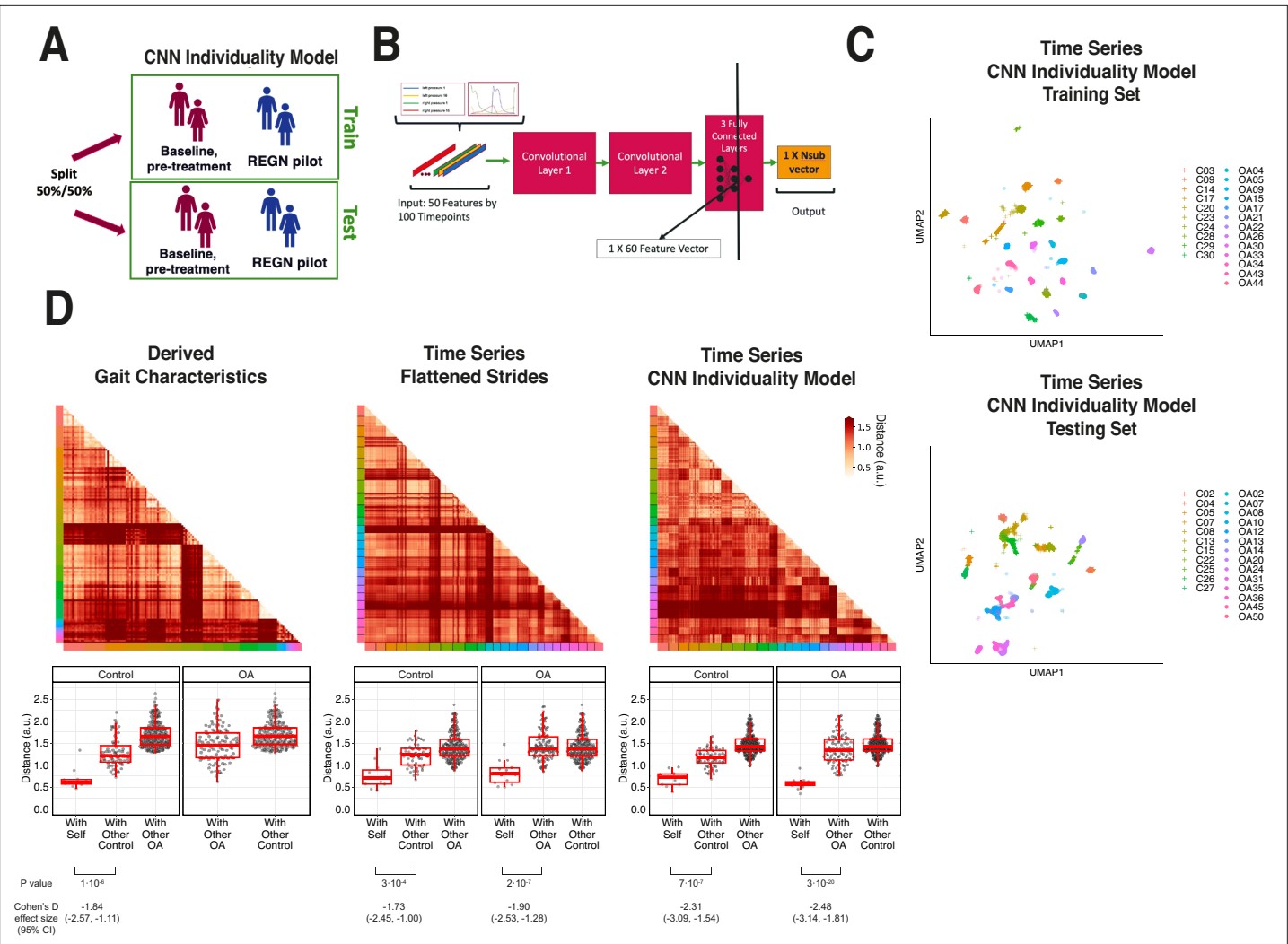

**Figure 5.** Latent convolutional neural net (CNN) representation of raw sensor time-series data from digital insoles: identifying subject-specific patterns of human gait. (**A**) Pilot study subjects and knee osteoarthritis (OA) clinical trial patients were split 50:50 into training and testing sets, stratified by disease status, for the first CNN model investigating the individuality of gait patterns. (**B**) A CNN was trained on segmented structured strides from the digital insole in the training set, to predict from which subject the stride came. The activation of the last fully connected layer in the CNN consists of 60 features and represents the model's latent representation of gait. (**C**) Uniform Manifold Approximation and Projection (UMAP) clustering of these 60 latent features for each stride captures the individuality of participants in both the training and testing sets. Each dot represents a single stride, colors represent each participant, and shapes represent participants' health status (C = control). Intra- and inter-subject clustering and separation is greater in the training set, as expected, and is present in the testing set as well. (**D**) Distances (in arbitrary units) between each pair of walks (for derived gait parameters) or strides (for time series) from the testing set shown as heatmaps for each of the three methods (top panels). Subject of the walk/stride are color identified along the edge. Boxplot of mean distance of each walk/stride with other walk/strides from the same individual, and with walk/strides from other individuals separated by disease class (bottom panels). Distances are faceted by the disease class of the individual. A good representation has low distance for 'with self', and high distance for 'with other' classes.

The online version of this article includes the following figure supplement(s) for figure 5:

**Figure supplement 1.** Example heatmap of a good representation that has low distance between all pairs of walks/strides from the same participant and high distance between all pairs of walks/strides from different participants.

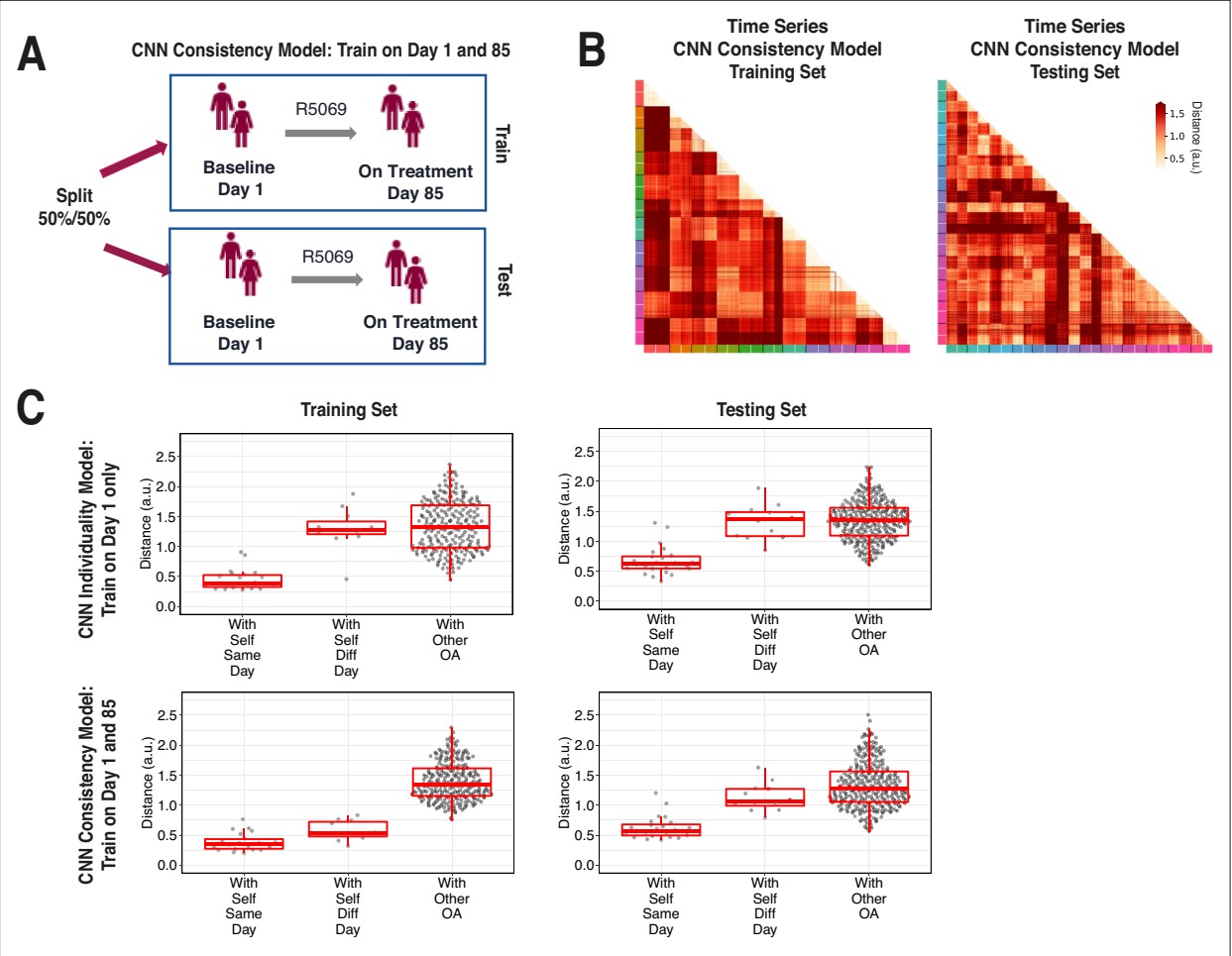

**Figure 6.** Training across multiple days increases consistency of convolutional neural network (CNN) model latent representation. (**A**) Knee osteoarthritis (OA) clinical trial participants were split 50:50 into training and testing sets containing both day 1 (baseline) and day 85 (on treatment) data, for the second CNN model investigating the consistency of gait patterns. (**B**) Distances (in arbitrary units) between pairs of strides in the latent representation from the consistency CNN model in the training and testing sets, shown as heatmaps. Strides from the same person are arranged next to each other, with strides from day 1 listed first then strides from day 85. Color along the edge indicates each person. (**C**) Boxplots of mean distance of each stride with other strides from the same person on the same day, from the same person on different days, and from other people, for both the individuality model (*Figure 5*) and consistency model (**A–B**). Distances are shown using the different models in both the training and testing sets.

The online version of this article includes the following figure supplement(s) for figure 6:

**Figure supplement 1.** Boxplots of mean distance (in arbitrary units) of each stride with other strides from the same person on different days for both the convolutional neural network (CNN) individuality model (*Figure 5*) and CNN consistency model (*Figure 6A and B*) in both the training and testing sets.

the distances of strides from different subjects from the same class, as measured by Cohen's d effect size, suggesting that the CNN latent representations best capture gait individuality, and as such was used for follow-up analysis.

## Evaluation of individual gait signatures after training on raw sensor time series from multiple days

A second CNN model was trained on combined data from both timepoints in the R5069-OA-1849 clinical trial, where input data was labeled only by participant and not by timepoint. We call the first model trained only on day 1 data an 'individuality' model, and the second model trained on both day 1 and day 85 a 'consistency' model (*Figure 6A*).

We tested both models on days 1 and 85 by evaluating the distance of the CNN penultimate layer of all strides with each other. *Figure 6B* plots the distance for the consistency model on both

the training (left) and testing set (right) participants. For each participant, day 1 and 85 strides are arranged next to each other, and the cross-day distance within each participant is shown in squares closest to the diagonal.

*Figure 6C* shows stride distances for within subject from the same day, within subject from different days, and within other subjects. Within the training set, the consistency model, which was trained on both day 1 and 85 strides, produced lower distances than the individuality model, which was trained on only day 1 strides, in comparing strides within self from different days (p<0.001, paired *t*-test) (*Figure 6—figure supplement 1*). Importantly, within the testing set, the consistency model produced lower distances than the individuality model in comparing strides within self from different days (nominal p=0.033, paired *t*-test), suggesting that training across multiple days improves the ability of the CNN consistency model to identify features that remain consistent across multiple visits. The results also show additional capacity for model improvement with respect to consistency of gait, as in the consistency model, the distances of the data to oneself were lower amongst strides from the same day versus different day in both training and testing participants (nominal p<0.001 and nominal p=0.004, respectively, *t*-test).

## Discussion

This study investigated whether gait data derived from a wearable digital insole appears to be accurate in comparison to a gold-standard laboratory reference technology (force plates) and whether it can answer questions of clinical interest (with the ultimate clinical goal of yielding useful endpoints). Overall, the digital insoles appeared to have face (analytical) validity for vGRF measurement in knee OA and controls (qualitatively appears to provide similar results to force plate data), and various insole-derived gait outcomes demonstrated early clinical validity as they provided clinically meaningful outcome measures for knee OA.

To address the question of how to estimate disease-specific patterns, we tested whether vGRF data allowed for detection of knee arthrophathy relative to control subjects with any device. We built an ML model using force plate vGRF data and tested the model on an independent force plate test dataset. We then extended the analysis to the data from the digital insole. We observed consistent disease versus control differences in vGRF curves with both technologies, such that these models were able to distinguish knee arthrophathy subjects from their respective control groups. The predictive performance of the ML model on data collected from a different device at different physical locations and experimental conditions suggests that the model was robust and generalizable (not overfit). While our analysis may be confounded by variables like age, our results imply that disease status is the major signal we observe. Our results suggest that vGRF data likely reflect true differences between control and knee arthrophathy subjects, suggesting that digital insoles may be used to screen for knee arthrophathy and potentially other diseases that impact gait.

We next sought to identify how different types of digital gait outcomes beyond vGRF (where the digital insole attempts to replicate exactly what is measured on a force plate) can be utilized. To provide additional insight into the advantages of utilizing wearable devices, we investigated alternative data types generated by wearable devices. Findings confirmed that speed-independent gait characteristics play an important role in identifying patients with knee arthrophathy and controls. Specifically, evaluation of derived gait characteristics from the digital insole highlighted walking speed as an important determinant of knee OA classification, which is expected (*Zeni and Higginson, 2009*); however, when derived gait characteristics highly correlated with speed were removed, the model still successfully detected knee OA subjects. Despite walking speed being a prognostic biomarker for mortality and a clinically meaningful outcome measure, it lacks the ability to detect disease-specific impairments or potential improvements with intervention (i.e., speed is the accumulation of other gait characteristics, and more specific gait characteristics can be mapped to specific underlying musculo-skeletal, neurological, or cardio-pulmonary impairment) (*Morris et al., 2016*; *Zhou et al., 2022*). This highlights that in addition to alterations in speed there are additional gait characteristics in knee OA that differentiate them from controls.

Distinguishing between control and disease subjects, where effect sizes are expected to be large, may have important clinical implications. However, it remains to be seen whether wearable devices can successfully detect disease severity where effect sizes are smaller. Our study was not designed to investigate smaller effect size differences, so future studies are needed to evaluate this further. If

future studies show this is feasible, this data would be important to determine disease progression or evaluate the impact of therapeutic interventions, and could have utility in clinical practice, as well as serve as an endpoint tool for clinical trials. A strength of using digital insoles over other lower cost technology (e.g., Inertial Measurement Units, or IMUs) is that they can accurately capture the gait cycle and weight transfer/bearing in OA patients, which was our primary clinical focus here. Unlike IMUs that require algorithms to estimate the initial and final contact of the foot (*Paraschiv-Ionescu et al., 2020*), digital insoles provide additional information such as the ability to determine when the foot is actually on the ground via raw pressure sensor data, thereby improving the accuracy of gait event detection and proceeding outcomes. This is the reason why digital insoles are often used for analytical validation of IMU algorithms in clinical populations (*Mazzà et al., 2021*). Although digital insoles may not be as hardwearing as IMUs (i.e., capacitors may deteriorate over time, so are not recommended for free-living assessment) (*Vu et al., 2020*), they are useful clinical tools for active mobility tasks in an OA population as they provide gait- and pressure-related metrics that are relevant for this clinical condition. Alternatively, in free living settings (e.g., passive monitoring with wearables over hours/days/weeks), digital insoles may have issues with device placement and sensor deterioration (*Vu et al., 2020*), unlike IMUs that have been shown to be comfortable and have high acceptance across various clinical populations (*Keogh et al., 2023*). The objectivity of the data also makes these suitable for endpoint tools. For example, derived gait characteristics could represent reliable registrational endpoints in clinical research, given that they describe objective aspects of gait with clinical relevance (e.g., total distance walked in meters, or maximum force applied during a 3 min walk in Newtons).

Finally, evaluation of raw time-series data from digital insoles demonstrated that data from a single stride could identify individual subjects. However, the interpretation of time-series data remains challenging, particularly when analyzed with deep learning methods. In current clinical trial settings, derived gait characteristics may be a simpler approach. Nevertheless, our finding that raw time-series data contain subject-specific latent features suggests that potentially useful gait features, beyond the disease signatures studied here, may exist in this data.

Collecting additional timepoints from individuals may permit the model to learn more consistent subject-specific gait patterns. Individual subject gait patterns have been reported previously (*Horst et al., 2019*; *Horst et al., 2016*; *Schöllhorn et al., 2002*), and understanding their quantification may be useful in clinical development for precision medicine applications. Subject-level gait patterns and the ability to identify unique signatures of an individual's gait may enable improved monitoring of treatment responses on a per-subject as well as on a population-wide level. Training datasets with participant data from multiple visits improve the ability of the model to detect gait features that remain consistent, or that change, with time. Collection and training on additional timepoints beyond the 2 d in our study may result in models that better learn gait features to consistently identify an individual across time.

This work highlights the potential utility of digital insoles for gait assessment in knee OA. However, the data such devices generate require clear hypothesis-driven validation to detect relevant signals just like research-grade instrumentation. Holistically, by showing that vGRF data from a digital insole replicate the standard clinical data generated from force plates, we demonstrated criterion validity of vGRF data from digital insoles, meaning that digital insoles can to some degree replicate a clinical standard (the criterion). We further demonstrated the external validity within the digital insole study of the disease gait signature across both methodologies (force plate and digital insole) using an ML approach, with a training set built entirely on force plate data and evaluated on both force plate and digital insole data collected elsewhere. Analytical strategies that maximize both clinical understanding and generalizability to other studies are of course standard for biomedical research. However, we go beyond this step, and further attempt an analytical approach to maximize construct validity—how close a digital biomarker reads out the 'construct' it is intended to measure—even at the expense of face validity (the degree to which a measure is intuitively interpretable), for a particular clinical question. Here, the fact that raw sensor data lacks face value interpretability but improves upon our ability to ascertain subject-specific gait patterns may inform us that these digital biomarker data contain disease or subject-specific information that could be leveraged in alternative clinical circumstances.

## Study limitations

Limitations of this analysis include the small sample size in the pilot study of controls, in addition to the fact that subjects in this study were not demographically and clinically matched with the OA study. Further, the OA study did not control for knee-only OA on one or both joints, thus making this population heterogeneous and limiting generalizability. Despite this, we were able to show favorable classification performance for these control subjects relative to their force plate counterparts. The study design also did not specifically account for familiarization with the insoles. While this study only looked at knee arthropathies, future work will be focused on a broader set of gait-affecting diseases. The study's analytical approach, which focused solely on regular walking patterns and excluded outliers and irregular patterns in the insole gait data, may limit the comprehensiveness of the findings; future research should aim to collect more diverse walking data per subject to include nonregular patterns. Additionally, the insoles are currently only capable of providing vGRF outcome measures, but medial and lateral GRF would also be useful outcomes for OA as these also relate to pain and OA severity (*Costello et al., 2021*). Technology developments would be required to develop a wearable system that would be capable of capturing comprehensive GRFs during walking in OA and should be an area of future research. Only two repeat timepoints were used for subject-level classification in the knee OA group, but we demonstrated a modeling approach to reliably compute an individual's gait consistency. In the future, this would ideally be performed on data collected at more than two timepoints. In addition, full analytical validation of the digital insoles was not possible as force plate and insole data was not collected simultaneously, and therefore evaluation of criterion and concurrent validity was not possible. Future work should help determine gait outcome measures for accuracy and error.

## Conclusions

This work outlines a framework for an integrated analysis of digital insole data to answer clinical and research questions relevant to digital biomarker development. To identify disease signatures, we built an ML model using only data from force plates, the clinical standard, and analyzing data from a digital insole, we showed comparable disease classification. This platform-agnostic analysis demonstrates that ML approaches can help support and validate digital biomarkers and may yield digital endpoints of clinical utility. In addition, the finding that our models can identify individual gait patterns suggests that data generated from a digital insole may have unforeseen future applications such as the potential to detect changes in individual gait patterns, which may provide better understanding of the impact of a therapeutic intervention for that individual. Ultimately, this work helps support the aspiration that digital technology may provide value in the healthcare delivery setting, aiding in accurate diagnosis or longitudinal monitoring of disease progression or of response to treatment.

## Methods

### Study design

The objectives were to characterize data from a wearable insole device (Moticon), demonstrate their utility relative to a clinical standard, and investigate optimal analytical methods and data types for the analysis relevant to clinical questions of interest. Three datasets were integrated for analysis (*Figure 1*). The GaitRec force plate vGRF dataset contained force plate control subjects (N = 211) and knee injury subjects (N = 625) (*Horsak et al., 2020*).

Volunteers were used as control subjects (N = 22) from a pilot study conducted between July 2019 and August 2019 to evaluate the usability of a digital insole to measure gait (*Table 2*). The date of first visit for the first volunteer in the pilot study was July 6, 2019, and last volunteer was August 5, 2019. Those pregnant or with a body mass index above 40 kg/m$^2$ were excluded from the study. Volunteers were recruited internally within the Regeneron facility located in Tarrytown, NY, and were provided consent prior to participation. The pilot study did not require IRB approval because the research was not subject to the Common Rule (45 CFR Sec 46.104) or FDA regulations and did not meet the definition of 'Human Research' under New York law.

As part of a clinical trial evaluating the impact of a novel pain therapeutic in moderate to severe knee OA (R5069-OA-1849; NCT03956550), a sub-study of the digital insole was performed to collect data for gait assessment in knee OA patients (results from the clinical trial are published separately) (*Somersan-Karakaya et al., 2023*). This information is available publicly in the protocol in Section

**Table 2.** Baseline characteristics and gait assessments of subjects in the digital insole pilot study and patients with knee osteoarthritis (OA) in the R5069-OA-1849 clinical trial digital insole sub-study. Note that this table represents the total subjects enrolled with data used in any analysis of this study. Specific Ns are given where relevant and reflect subsets of these subjects.

| | Controls Pilot study (N=22) Cross-sectional | Knee osteoarthritis Clinical trial (N=44 enrolled in sub-study, N=43 data collected) Longitudinal |
|---|---|---|
| Age (years) | | |
| Mean | 39 | 62.75 |
| Median | 35 | 63 |
| Range | 19–85 | 52–77 |
| Sex | | |
| Female | 11 | 28 |
| Male | 11 | 8 |
| Body Mass Index (kg/m²) | | |
| Mean | 26.2 | 34.5 |
| Median | 26.0 | 34.8 |
| Range | 20.1–37.0 | 26.6–38.9 |
| Arthropathy class | N/A | K-L2-3: N=23 K-L4: N=13 |
| Walk test | Walk straight for 30 s | 3 min walk test (3MWT) |
| Walking speed | 3 speeds (comfortable, fast, slow) | 1 speed (comfortable) |
| Number of walk test performed | ~12 times (at each speed) | 1 time at baseline and 1-time on-treatment (day 85) |
| Total length of gait evaluation | 20 min | 3 min |

K-L, Kellgren–Lawrence.

8.2.6.6, Moticon Digital Insole Device Sub-Study for Gait Assessments. All patients in this sub-study were enrolled at two study sites in the United States and Moldova, and the study was conducted between June 2019 and October 2020. The date of first enrollment in the R5069-OA-1849 trial June 17, 2019, and last patient visit was October 29, 2020. The sub-study targeted to enroll approximately 13 patients per treatment group to obtain data on at least 10 patients per treatment group for a total of approximately 30 patients across the entire sub-study. The treatment groups were as follows: patients were randomized in a 1:1:1 ratio to receive a low dose of REGN5069 at 100 mg IV every 4 weeks (Q4W), and a high dose of REGN5069 at 1000 mg IV Q4W, or matching placebo Q4W. Eligible participants were men and women ≥40 years of age with a clinical diagnosis of OA of the knee based on the American College of Rheumatology criteria with radiological evidence of OA (Kellgren–Lawrence score ≥ 2) at the index knee joint as well as pain score of ≥4 in Western Ontario and McMaster Universities Osteoarthritis Index (WOMAC) pain sub-scale score. The WOMAC score is a self-administered questionnaire consisting of 24 items divided into three subscales, where the pain sub-score is assessed during walking, using stairs, in bed, sitting or lying, and standing upright. The study protocol received Institutional Review Board and ethics committee approvals from Moldova Medicines and Medical Device Agency and National Ethics Committee for Moldova, and the Western Institutional Review Board.

## Study procedures

In the pilot study, each participant walked straight along a hallway with a hard tile floor at three different qualitative speeds for ~12 times at each speed (~36 walks total). For each walking trial,

participants wore the digital insole inside their own shoes and were prompted to walk at a normal or comfortable speed, walk fast as if they were in a hurry (fast speed), or walk slow as if they were at leisure (slow speed). Prior to the walking trials, each participant was instructed to practice walking around to get accustomed to the insole. Participants' clinical and demographic information was also collected prior to walking trials.

In the R5069-OA-1849 clinical trial, a total of N = 44 OA patients were enrolled into a sub-study of a 259-patient clinical trial. Patients were required to bring the same pair of shoes to the study site to perform a 3MWT with the digital insole. Each patient performed the task twice, once at baseline and the other 85 d later post-treatment.

## Equipment

Moticon digital insoles (Moticon Rego AG, Munich, Germany) were used to derive wearable gait outcomes from the participants. Each Moticon digital insole has a total of 25 sensors per foot: 16 vertical plantar pressure sensors that assess force, a trial-axial accelerometer that measures acceleration, and a gyroscope that measures orientation and angular velocity. Each sensor captures data at 100Hz, and dedicated software computes several clinically relevant spatial and temporal-derived gait characteristics comparable to data generated in a gait lab. The Moticon digital insole computes vGRF in the same way as a force plate, generating comparable data outputs. In addition to vGRF data, derived gait characteristics summarizing a subject's walk and raw sensor time-series data can be obtained.

## vGRF data processing

To normalize the vGRF data across devices (due to the differing sampling frequencies of force plates and Moticon) and subjects, smoothing spline functions (scipy.interpolate.interp1d) were fit to vGRF time-series sensor data from both GaitRec force plate data and Moticon-computed vGRF data. vGRF curves were bounded by 0, and 100 evenly spaced timepoints across the curve were derived for each curve (to derive a % stance phase). All vGRF curves were normalized by participants' body weight in Newtons. Within each device, the vGRF curves were further normalized using a z-transformation within each stance phase timepoint (*Figure 2A*).

## Linear models to associate covaraites with vGRF signal

Using the GaitRec force plate dataset, consecutive linear models were fit at each of the 98% stance phase timepoints. We used disease (knee arthropathy or control), age, sex (male or female), and body weight as covariates in the model (linear model (lm) and ANOVA function, R), with each subsequent vGRF % stance phase timepoint as the dependent variable. Within each linear model, using the sum of squares for each category divided by the total sum of squares, we calculated the variance of each component's contribution to the total variance, with the residuals indicating the unexplained variance in these models. The use of linear models at each of the 98 points during the % stance phase allowed us to examine the relationship between vGRF and the covariates (disease, age, sex, and body weight) at each specific point in time during the stance phase of walking. This is important as the relationship between these variables and vGRF may change throughout the stance phase.

## Digital insole raw sensor time-series data processing

The digital insole collects 25 100-Hz measurements for each foot (50 measurements across both feet), comprising 16 measurements from 16 vertical plantar pressure sensors, 3 x,y,z measurements from an accelerometer, 3 x,y,z measurements from a gyroscope, 1 measurement of total force, and 2 x%,y% measurements of center of pressure. These raw sensor time-series sensor data for both the R5069-OA-1849 clinical study and Regeneron pilot study was preprocessed with custom scripts written in Python 3.6.

For the following analysis, a 'walk' was defined as data captured by the digital insole while the subject completed the researcher's walking task (typical duration of 180 s for the R5069-OA-1849 clinical study and 25 s for the Regeneron pilot study). A 'stride' is defined as the data captured by the digital insole between the peak pressure of the right heel (the average of digital insole right pressure 1 and 2 sensors) and the next peak pressure of the right heel. A typical stride duration is 1–2 s, highly dependent on individual walking speed.

Data preprocessing for each subject was performed separately. First, each walk was segmented into individual strides. Since the digital insole did not collect data in regular intervals, each stride was interpolated for each of the 50 sensors to obtain 100 timepoints along the stride. Thus, each interpolated stride consists of 50 vectors (one for each sensor), and each vector is 100 units long.

For the pilot study, walks from each subject were processed individually, treating slow, comfortable, and fast walks separately. Walks without all 50 features and walks with greater than 5% missing data were excluded. For the remaining walks, any missing data was linearly interpolated (scipy.interpolate.interp1d).

Each walk was then segmented into strides, and each stride was interpolated to 100 timepoints. To segment a walk, peaks were identified in the average time series of the Moticon right pressure sensors 1 and 2, located in the right heel using scipy.signal.find_peaks with parameters width = 10 and prominence = 5. The walk was segmented using the peaks, and the number of measurements in each segment was calculated. Segments that had that an outlier number of samples (outliers defined as 1.5 * iqr ± q3 or q1) were excluded, such that only regularly repeating segments, or strides, were analyzed. Each of the 50 features in each stride was then linearly interpolated (scipy.interpolate.interp1d) to 100 time points. Only walks with at least 10 interpolated strides were further analyzed.

Under the assumption that an individual's strides within a walk should be highly regular to each other, each stride's Pearson r correlation with the means of the remaining strides was computed (stats.pearsonr), and any strides with an outlier Pearson r correlation (outliers defined as 1.5 * iqr ± q3 or q1) were excluded. This process was then repeated with the remaining strides to obtain a list of the Pearson r coefficients of each stride with the means of the other strides. The entire walk was excluded if the mean of the Pearson r coefficients fell below 0.9. This procedure was repeated one last time, across all walks by an individual at the same walking speed (slow, comfortable, fast). That is, each stride's Pearson r correlation with the average of remaining strides in all walks at the same speed was computed. Again, assuming strides within a subject and within a given walking speed should be consistent with each other, strides with an outlier Pearson r correlation were excluded (outliers defined as 1.5 * iqr ± q3 or q1). Lastly, features dependent on body weight (i.e., pressure sensors and force sensors) were normalized by the subject's mass.

For the R5069-OA-1849 clinical trial, data were processed similarly. OA patients had digital insole data collected for only two walks, on day 1 and on day 85, which were processed separately.

## Derived gait characteristics and identification of those speed-correlated

The digital insole derives 85 gait parameters from each walk. Of those, three are directly related to the length of the walk (walking distance and left and right center-of-pressure trace length) and were excluded from further analysis, leaving 82 derived gait characteristics.

Spearman correlations between these 82 parameters were calculated across all walking speeds (slow, comfortable, fast). The silhouette method was used to determine the optimal number of clusters with the factoextra package in R with function fviz_nbclust with 100 bootstrapped samples. Since we were interested in understanding aspects of gait other than walking speed, we correlated all parameters against walking speed and conservatively removed 14 parameters that may be influenced by walking speed in any way (|Spearman rho| > 0.7). This allowed for an investigation into gait parameters less influenced by walking speed.

## Dimensionality reduction

UMAP method for dimensionality reduction was applied using the R UMAP package with default parameters to the z-transformed vGRF data from both the force plate and digital insole datasets to investigate batch effects.

PCA of digital insole vGRF, derived gait characteristics, and raw sensor time series was performed using the prcomp function in the R stats package. Heatmaps of Moticon parameters are displayed per individual, averaged across all individual walks. All heatmaps displayed derived gait characteristics after z-transformation by row across all subjects. All clustering on heatmaps was unsupervised, within groups.

## ML model building

SVM models were built using vGRF, derived gait characteristics, and raw sensor time-series processed data using the sklearn package in Python. This model choice was also benchmarked against logistic regression and XGBoost models from the sklearn and xgboost packages in Python (*Figure 2—figure supplement 4*).

The force plate dataset was randomly split into 85% training and 15% hold-out test datasets. The 85% training dataset was used for LOOCV and to construct a final trained model, which was then evaluated on the hold-out test dataset. The digital insole dataset was also used as an independent dataset for evaluating the model.

Model performance was evaluated using multiple methods. ROC and precision-recall curves were used to evaluate overall performance. Additionally, we quantitated the auROC curve, which is a standard measure of classification success and describes model performance regardless of baseline likelihood for either class. In addition, we quantitated the auPR and F1-scores, which are useful for evaluating datasets with class imbalances.

## Subject-specific gait signatures

A second question we sought to investigate is whether we could determine subject-specific gait signatures. If we could train models to identify individual subjects from their walk, or from just a single stride, this may suggest that the gait data collected has at a minimum that ability to identify attributes beyond knee disease. We posed this question irrespective of disease state and rather focused on identifying the optimal method to determine an individual participant's gait pattern.

We approached this question via two approaches commonly used in clinical research settings (*Horst et al., 2019*; *Chau, 2001a*; *Chau, 2001b*; *Schöllhorn, 2004*). The first is regarding the individuality of human gait patterns. We sought to understand which methodology is best suited to identify generalizable patterns of any person's gait. The second is understanding which methodology captures features of a specific individual's gait that have consistency with time. By analogy, a facial recognition software should identify people regardless of whether they wear hats or sunglasses. To learn that these accessories are not stable features of an individual, ML models would need to be trained on images of the same individuals with and without such accessories; that is, trained on these individuals across multiple timepoints. Similarly, we sought to understand whether training on the same individuals across multiple timepoints improves the ability of our models to detect features that identify individuals consistently with time.

## CNN model for control versus OA classification

For the control versus OA model, model performance was determined using LOOCV. A CNN was trained using all strides from all but one participant, after which the model was evaluated on all strides of that left-out participant. Each participant was used as a left-out test participant in one model, such that for $N$ participants, there were $N$ different CNN models each trained on the other $N$-1 participants. Each stride was labeled as to whether it came from a control or an OA participant.

For each CNN model, strides from the $N$-1 training participants were split into an 80% training set and a 20% validation set. Each feature within each stride was scaled into 0 min to 1-max range across the 100 interpolated timepoints. The normalized data from each stride was then used as the input to the following CNN architecture: (Functions in italics from the Python [v3.9.7] PyTorch [v1.8.0.post3] package torch.nn were used with the default parameters unless otherwise noted.)

- First 1D convolution layer with 50 in channels, 64 out channels, and a convoluting kernel of size 3 (*Conv1d*).
- Element-wise rectified linear activation unit (*relu* in torch.nn.functional).
- 1D max pooling with a sliding window kernel of size 2 (*MaxPool1d*).
- Dropout with 0.2 probability (*Dropout*).
- Second 1D convolution layer with 64 in channels, 128 out channels, and a convoluting kernel of size 3.
- Element-wise rectified linear activation unit.
- 1D max pooling with a sliding window kernel of size 2.
- Dropout with 0.2 probability.
- Flatten data to a linear vector of 2944 elements.
- First fully connected layer with 2944 in channels and 120 out channels (*Linear*).

- Element-wise rectified linear activation unit.
- Dropout with 0.2 probability.
- Second fully connected layer with 120 in channels and 32 out channels.
- Element-wise rectified linear activation unit.
- Dropout with 0.2 probability.
- Third fully connected layer with 32 in channels and 1 out channel.
- Logistic sigmoid function (*sigmoid* in torch).

Binary cross entropy loss (*BCELoss*) was used as the loss function, and stochastic gradient decent (*SGD* in torch.optim) with a learning rate of 0.001 and momentum of 0.9 was used as the optimizer. Data was loaded into the CNN in batches of 32 with shuffling (*DataLoader* in torch.utils.data), and backward propagation and parameter optimization were conducted in such batches. Models were trained for 10 epochs, and model parameters from the epoch with the best accuracy on the validation set were chosen as the final model parameters. The model was then tested on the strides of the left-out participant. Model predictions for whether each stride from the left-out participant was from a control or an OA participant were aggregated across the *N* CNN models, and the overall classification performance was computed.

## CNN model for subject classification and latent representation

As the digital insoles produced high-frequency raw sensor time-series data, we analyzed whether such structured strides (50 measurements along 100 interpolated timepoints for each stride) contained informative subject-specific gait features. To utilize the temporal aspect of the data, we constructed a one-dimensional CNN in which the model could interpret the relationship between sequential time-points for each sensor. This temporal relationship in the input data was lost in our previous analysis, in which the stride was flattened and interpreted by SVM.

For the individuality and consistency CNN models, the model was trained to identify the subject from which a stride came. However, the purpose of using the CNN model was not to classify training subjects based on their strides, but rather to extract activation of the penultimate fully connected layer for the model's latent representation of the 'gait fingerprint' of a stride. As such, once the CNN model was trained on participants in the training set, the model was then applied to participants in the hold-out testing set and latent representations for each stride were extracted.

To train the CNN model, strides from the training participants were split into a 64% training set, a 16% validation set, and a 20% final validation set. A similar CNN architecture was used as before, except now rather than a binarized control versus OA output, the model outputs the subject label. As such, the model architecture differed starting from the second fully connected layer:

- Second fully connected layer with 120 in channels and 60 out channels.
- Element-wise rectified linear activation unit.
- Dropout with 0.2 probability.
- Third fully connected layer with 60 in channels and 23 out channels.

The CNN model was trained in the same manner as before, except multi-class cross entropy loss (*CrossEntropyLoss*) was used as the loss function. As before, the model was trained for 10 epochs, and model parameters from the epoch with the best accuracy on the validation set were chosen as the final model parameters. The final validation set was then used to check the final model's performance. A forward hook (*register_forward_hook* in torch.nn.modules) was attached to the penultimate fully connected layer, to extract activation of that 60-element layer for a new stride inputted into the model.

## Evaluation of subject individuality across different representations

Models were constructed for each data type to predict individual subjects in the training set and then applied on the testing set. Next, distances between each pair of walks/strides were calculated within a subject, within other subjects with the same disease status, and within other subjects with a different disease status.

Each feature was first z-scored (centered and scaled to unit variance, using *scale* function in base R), and Euclidean distances between all walks/strides in the testing set were calculated using *dist* function in the R stats package. To compare across representations with differing number of features, distances

were divided by the square root of the number of features. The mean distance between every two participants (including with oneself) was then calculated.

To evaluate models for subject individuality, each participant-to-participant comparison was categorized into the groups of control within-self, OA within-self, control with another control, OA with another OA, or one control with one OA. Significance of difference in distances between participant categories was analyzed with *t*-tests in the R stats package. Effect sizes as Cohen's d were computed with the R effsize package. Throughout, assumptions of the *t*-test were checked through creation of univariate histograms of each variable to qualitatively test for normality (i.e., does the data look Gaussian).

## Evaluation of CNN models of subject individuality and consistency

Digital insole sensor data from both baseline (day 1) and on-treatment timepoints (day 85) of OA participants in the R5069-OA-1849 clinical trial was used to evaluate whether training on data from 2 d instead of just 1 d improves the consistency of the CNN representation of participants. A second consistency CNN model was trained on combined data from both timepoints for training set participants, where input data was labeled only by participant identity and not by timepoint. For comparability, both the individuality and consistency CNN models used the same split of train and test participants. Both models were given day 1 and day 85 of testing set participants, and the distance between all stride pairs as represented by the penultimate CNN layer was calculated as before.

To evaluate models for consistency, each participant-to-participant comparison was then categorized into the groups of within-self same day, within-self different day, or subject with another subject. Only OA participants were analyzed for consistency as only they were assessed on two different days. Significance of difference in distances between the CNN individuality and consistency models across the same participant comparisons was analyzed with paired *t*-tests in the R stats package.

## Acknowledgements

We would like to give a special thanks to the R5069-OA-1849 clinical trial participants who volunteered for the Moticon digital insole sub-study. We also thank Mark Waterlow, Scientific Director at Prime, for writing, formatting, and editorial assistance; Qing Zhou and Richa Attre in Regeneron Medical Affairs for editorial and project management support; and Daniel Choka from the Regeneron Creative Services Department for assistance with creating the figures. We would also like to give a special thanks to Dr.-Ing. Robert Vilzmann, CTO, Moticon ReGo AG, for helpful discussions throughout the development of this work.

## Additional information

### Competing interests

Matthew F Wipperman, Allen Z Lin, Kaitlyn M Gayvert, Benjamin Lahner, Selin Somersan-Karakaya, Xuefang Wu, Joseph Im, Minji Lee, Bharatkumar Koyani, Ian Setliff, Malika Thakur, Daoyu Duan, Aurora Breazna, Fang Wang, Wei Keat Lim, Gabor Halasz, Jacek Urbanek, Yamini Patel, Gurinder S Atwal, Jennifer D Hamilton, Samuel Stuart, Oren Levy, Andreja Avbersek, Rinol Alaj, Sara C Hamon, Olivier Harari: Employee and shareholder of Regeneron Pharmaceuticals, Inc.

## Funding

| Funder | Grant reference number | Author |
|---|---|---|
| Regeneron Pharmaceuticals | | Matthew F Wipperman<br>Allen Z Lin<br>Kaitlyn M Gayvert<br>Benjamin Lahner<br>Selin Somersan-Karakaya<br>Xuefang Wu<br>Joseph Im<br>Minji Lee<br>Bharatkumar Koyani<br>Ian Setliff<br>Malika Thakur<br>Daoyu Duan<br>Aurora Breazna<br>Fang Wang<br>Wei Keat Lim<br>Gabor Halasz<br>Jacek Urbanek<br>Yamini Patel<br>Gurinder S Atwal<br>Jennifer D Hamilton<br>Oren Levy<br>Andreja Avbersek<br>Rinol Alaj<br>Sara C Hamon<br>Olivier Harari |

The funders had no role in study design, data collection and interpretation, or the decision to submit the work for publication.

## Author contributions

Matthew F Wipperman, Conceptualization, Data curation, Software, Formal analysis, Supervision, Investigation, Visualization, Methodology, Writing – original draft, Project administration; Allen Z Lin, Conceptualization, Data curation, Software, Formal analysis, Supervision, Investigation, Visualization, Methodology, Writing – original draft; Kaitlyn M Gayvert, Conceptualization, Data curation, Software, Formal analysis, Investigation, Visualization, Methodology, Writing – original draft; Benjamin Lahner, Conceptualization, Software, Formal analysis, Investigation, Methodology; Selin Somersan-Karakaya, Supervision, Investigation, Writing – original draft; Xuefang Wu, Conceptualization, Data curation, Supervision, Methodology; Joseph Im, Conceptualization, Investigation; Minji Lee, Formal analysis; Bharatkumar Koyani, Supervision, Project administration, Writing – review and editing; Ian Setliff, Malika Thakur, Daoyu Duan, Formal analysis, Methodology; Aurora Breazna, Data curation; Fang Wang, Data curation, Formal analysis; Wei Keat Lim, Gurinder S Atwal, Supervision, Methodology; Gabor Halasz, Supervision, Investigation; Jacek Urbanek, Methodology, Writing – review and editing; Yamini Patel, Investigation; Jennifer D Hamilton, Supervision, Writing – review and editing; Samuel Stuart, Writing – review and editing; Oren Levy, Andreja Avbersek, Supervision, Writing – original draft; Rinol Alaj, Conceptualization, Supervision, Investigation, Project administration, Writing – review and editing; Sara C Hamon, Conceptualization, Data curation, Formal analysis, Supervision, Investigation, Methodology, Writing – review and editing; Olivier Harari, Conceptualization, Formal analysis, Supervision, Investigation, Methodology, Project administration, Writing – review and editing

## Author ORCIDs

Matthew F Wipperman ⓘ https://orcid.org/0000-0003-1436-3366
Allen Z Lin ⓘ http://orcid.org/0000-0001-5105-308X
Kaitlyn M Gayvert ⓘ http://orcid.org/0000-0001-9135-6501
Wei Keat Lim ⓘ https://orcid.org/0000-0002-6226-2570
Olivier Harari ⓘ http://orcid.org/0000-0001-8214-489X

## Ethics

Clinical trial registration NCT03956550.

For the pilot study, participants were recruited internally within the Regeneron facility located in Tarrytown, NY, USA, and were provided written informed consent prior to participation. The study was considered exempt research under the Common Rule (45 CFR Sec 46.104). The R5069-OA-1849

study protocol received Institutional Review Board and ethics committee approvals from Moldova Medicines and Medical Device Agency and National Ethics Committee for Moldova, and the Western Institutional Review Board.

## Decision letter and Author response
Decision letter https://doi.org/10.7554/eLife.86132.sa1
Author response https://doi.org/10.7554/eLife.86132.sa2

## Additional files

### Supplementary files
- MDAR checklist
- Supplementary file 1.

### Data availability
All relevant demographic and clinical information, all vGRF, derived gait characteristics, and raw sensor time series data, in addition to R and Python scripts used to perform the analysis have been uploaded here: https://github.com/regeneron-mpds/wipperman_digital_wearable_insole_ML (copy archived at *REGN-MPDS, 2024*). The GaitRec dataset is available online here: https://doi.org/10.1038/s41597-020-0481-z.

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
