## [Editor Report]

This study presents a valuable dataset and tool that can aid in arthropathies' assessment, potentially enabling such evaluation to be done outside the lab. There is convincing evidence supporting the comparison between the force plate and insole data but the evidence for distinguishing disease signatures is inconclusive and would need further development. This work will be of interest to physical therapists, clinicians, and researchers in the field of lower limb joint diseases.

---

## [Decision Letter]

**Decision letter after peer review:**

Thank you for submitting your article "Analysis of a digital insole versus clinical standard gait assessments in participants with knee arthropathy and health controls: development of a machine learning model" for consideration by *eLife*. Your article has been reviewed by 3 peer reviewers, one of whom is a member of our Board of Reviewing Editors, and the evaluation has been overseen by Tamar Makin as the Senior Editor.

The reviewers have discussed their reviews with one another, and the Reviewing Editor has drafted this to help you prepare a revised submission. We kindly request your attention to the following essential revisions. Additionally, we would greatly appreciate it if you could review the complete feedback and consider addressing the remaining concerns, if feasible.

Essential revisions (for the authors):

1) There is a need for improved cross-validation. Specifically, clarify the number of cross-validation folds and method for data splits (subject-wise or across the whole dataset) for the XGBoost model, reporting both the mean and standard deviation for the cross-validation split. Furthermore, for the CNN, leave-one-out cross-validation across 44 participants, results in only 2.3% of the data used for a test set, which can very likely result in overfitting, thus needs to be more stringently tested for this.

2) Explicitly state how much data were discarded due to missing signal from one or more sensors or to outliers and regularity of repeated patterns (address concerns that only regular patterns were used, how this can skew the data, and why it was not done for the force plate data). Furthermore, why was the data from the clinical trial processed differently (different exclusion criteria for percentage of missing data and Pearson r coefficient threshold)?

3) If possible, quantitatively compare the simultaneous recordings of the two systems (force plates and pressure insoles) or amend the statements regarding such comparison to reflect the concern and state that it is only qualitative.

4) State whether a correction for multiple comparisons was applied to the t-tests, and if it is not included, it should be applied and the new results should be reported.

5) There is a need for an additional data analysis regarding the specification of the severity or type of gait impairment (can you classify different severity and discriminate between affected joints). This is vital when there are claims about disease signatures, or disease classifications, as otherwise, the results can reflect general cautious, slow, or antalgic gait rather than the disease.

6) There is a need for further evidence against the possibility that the only thing that the model classifies is slow vs normal pace walking. For example, can a simple model trained on walking speed compare in performance to the current results?

7) Compare the deep learning models developed here to the performance of simpler models like linear regression and support vector machine.

8) Provide further information and manuscript changes in accordance with the reviewers' comments.

*Reviewer #1 (Recommendations for the authors):*

(1) The made equivalency of ground reaction forces (GRFs) obtained via force platforms (load cells or force transducers) and pressure insoles (pressure sensors) can be misleading (e.g. line 415). The same applies to comparing the abilities of the pressure insoles to those of force platforms by using one (vertical GRF – vGRF) of the multiple variables (3-axial forces and moments as well as centre of pressure) to compare their performance, while in some cases (raw sensor data and derived gait characteristics) utilizing all of the data from the pressure insoles. Although it can be understandable to use vGRF to compare to the one from pressure insoles (as this is the most consistent variable obtained with pressure insoles), it can lead the readers to believe that the system equivalence is apt. The statements equating measurements from these two systems and stating the outcomes of such analysis should be amended to reflect that.

2) Similarly, the comparison of vGRF between force platforms and pressure insoles is only qualitative (demonstrating that the two measurements look similar in the control case) and even then, the pattern for knee OA is significantly different from the one of knee arthropathies demonstrated from the force platforms. If possible, can simultaneous recordings of the two systems (force plates and pressure insoles) be quantitively compared, or if such data does not exist in the dataset, amend the statements regarding the comparison to reflect that?

3) Digital insole data pre-processing:

a. In the Methods Section, it is stated that since the digital insoles do not collect data in regular intervals the data was interpolated. This is confusing since data collection intervals should not matter as the analysis is not done in real-time. Can you clarify what is meant by this? Does it mean that the data is sampled at 100 Hz but these 100 samples per second are not regular (i.e. the first 100ms contain 20 samples while the next 100ms contain 5 samples) or is it meant that the data was up-sampled to 1000 Hz from 100 Hz to match the force plate data sampling rate? Also, state the force plate sampling rate and the brand/model of the equipment.

b. Further in the section, it states that walks with missing data from 1 or more sensors, or with missing data from 1% of the walk were discarded. Can you state how much data was discarded as a percentage of the overall data?

c. Similarly, data was discarded not only based on outliers but also on the regularity of repeated patterns. How much of the data was discarded as a percentage of the overall data in this part? Also, this is skewing the data to consider only regular patterns, which should be explicitly stated when comparing results and discussing them as it is a vital detail and was also not a thing done for the force plate data.

d. Why was the data from the clinical trial processed differently (different exclusion criteria for percentage of missing data and Pearson r coefficient threshold)? These have to be the same no matter what for comparative reasons.

e. Can you clarify more on the treatment population groups and the number of participants? Line 587 states 13 patients per treatment group aiming for a total of 30 patients, but the different treatment groups were not stated.

4) Can you comment on the pressure insoles size and fit for the participants, were there any cables connecting the insole leads to devices on other parts of the body, were the fit and the presence of insoles affecting the normal gait of the participants?

5) Concerning the derived gait parameters – It is explicitly mentioned in the Methods Section that the investigated gait parameters were less influenced by walking speed, but the Results section states that walking speed correlates were excluded rather than that they were significantly reduced. It is an important distinction, especially since the threshold was "Spearman rho > 0.7".

6) Lack of model cross-validation folds information:

a. For the XGBoost model – the force plate dataset was randomly split into 85% training and 15% testing datasets. Which force plate dataset was used (GaitRec?) and, more importantly, how many times was this repeated (how many cross-validation folds)? Was the split done across participants (85% of participants for training and 15% for testing) or across the whole dataset? If it is for the whole dataset, can you report a participant split? This will indicate how general the model is and testify to its strength to be used in a typical scenario when you want to predict OA in new patients.

The mean and standard deviation (SD) for the cross-validation needs to be reported. Furthermore, this trained model was further tested using the R5069-OA-1849 clinical study dataset only, is that correct?

b. Similarly, leave-one-out cross-validation (LOOCV) was applied for the convolutional neural network (CNN) model but there are no mentions of how many folds. How many participants, in total, was this model trained on and did you cycle over all of them (number of folds = number of participants) or was this done only once? If done only once, it needs to be repeated for all participants and the results for the mean and SD from these need to be reported.

c. Furthermore, the CNN results need to be tested for overfitting. For example, if the LOOCV of the CNN included 44 participants, then 1 participant represents only ~2.3 % of the data, meaning that the model most likely overfits this dataset. Thus, this needs to be checked by including more participants in the cross-validation (i.e. 85%/15% split across participants). This is raised since the CNN performance for the classification of knee OA using derived gait characteristics achieves auROC = 1 and auPR = 0.999, which can be strongly indicative of overfitting and needs to be checked.

d. Why was the cross-validation split different for the individuality and consistency CNN models and why is the last output consisting of a 23-element vector (shouldn't that be 44 to reflect the 44 different participants from the R5069-OA-1849 clinical study?)?

7) The results for subject-specific gait signatures are very exciting. However, the further claims on the ability to identify clinical attributes beyond knee disease as well as generalizability of the method are not substantiated. The results demonstrate that if data is included from both days of recording (day 1 and day 85 of treatment), there is a trend (P=0.033) suggesting it could be possible to identify the subject from others even across days. However, for this to work, you need training data from both timepoints of treatment, which is not the overall aimed application. It is still great that the distinction between individual participants is preserved regardless of the treatment time point! Although, the statement for the generalization across days has to be amended to reflect the results as the results from training on day 1 only reflect that it cannot distinguish between the day 85 data from the same participant and the rest. Similarly, the claim for a richness of data to help identify other clinical attributes is not corroborated by these results since they indicate that one is able to find different classification boundaries between participants. This does not necessarily reflect differences in clinical attributes as they could be solely based on different walking patterns and symmetry in pressure distribution.

8) Statistics – was there a correction for multiple comparisons for the t-tests? If so, that should be stated in the methods and, if not, it should be included and reported. Can you also overlay statistical significance on the plots?

9) Can you include comments on the additional benefit of using pressure insoles instead of inertial measurement units (IMUs) and include classification results using just the acceleration and angular velocity data? In the manuscript, it is mentioned a few times that gait speed (a feature that can be derived from a much cheaper IMU) is one of the main features to help classify OA. The further analysis in the paper demonstrates that features that are less correlated with speed can achieve a better performance, which is great. However, it will be good to quantify explicitly the IMU role and clarify further what else could be possible by having the additional pressure data, which can excite the reader even more and give further perspective.

10) Similarly, can you elaborate more on the importance of identifying participants using the CNN approach investigated in this work? There is a sentence about this on line 405 but it would be beneficial for the reader to understand more and will show the importance of your results.

11) The inclusion of the foot as an important dependable for classification in the XGBoost model (line 269) should be stated to be because the OA participants were injured only unilaterally (left leg), thus, this will generate these differences regardless and should be put as a confound. Have you tried classifying only from the "healthy" leg in OA participants, to demonstrate that the affected side does not matter?

12) Can you comment on the fact that knee OA participants are walking more slowly, which is indicated by the clustered data for vGRF and less by the gait characteristics, but the raw sensor data clusters the OA right next to the fast walking control group (Figure 4 A-C)? What causes this different and counterintuitive clustering?

13) Can you elaborate on the linear model fitted to each point along the vGRF curve (visualization of knee arthropathy signatures from vGRF data, Supplementary Figure 2) in the Methods Section?

*Reviewer #3 (Recommendations for the authors):*

– Insoles have been used for gait assessment in multiple groups, and validation studies exist. The introduction might benefit from a summary of these findings.

– In the introduction, a rationale could be provided for why knee arthropathy was chosen for the current paper. In the eyes of the authors, what does this group reflect? Please also discuss previous studies regarding GRF in knee OA (e.g. Costello et al., Osteoarthr Cartilage 2021; Costello et al., Br J Sports Med 2023).

– In the methods, I missed information about which knee injuries were included. In line with my previous comment, did they all have pain above a certain threshold? This group can be very heterogeneous.

– The set-up of having only one group with disease vs controls, is severely limiting the potential impact of the ML model. If the authors would be able to test if this model could be trained to differentiate between pain levels, or which joints are involved, that would substantially improve the impact. See for example Leporace et al. Clin Biomech 2021.; Kushioka et al. Sensors 2022.

[Editors' note: further revisions were suggested prior to acceptance, as described below.]

Thank you for resubmitting your work entitled "Digital wearable insole-based identification of knee arthropathies and gait signatures using machine learning" for further consideration by *eLife*. Your revised article has been evaluated by Tamar Makin (Senior Editor) and a Reviewing Editor.

After detailed discussions with our review panel, it is clear that significant concerns remain unaddressed. These issues are critical for the integrity and publication viability of your work.

Key areas requiring your immediate attention include:

1. Methodological Rigour: There are serious concerns about your choice of cross-validation method and modelling approach, given the heterogeneity of the OA population. A comparative analysis with simpler models is also lacking, raising questions about overfitting.

2. Data Presentation and Analysis: Discrepancies in data handling and presentation, particularly in Figure 4 and your exclusion criteria, are worrying. These issues cast doubts on the validity of your comparisons and findings.

3. Comprehensive Explanation: Essential details are missing or inadequately explained. This includes the need for clearer elaboration on the linear model in the Methods Section and a more thorough discussion on the limitations and assumptions of your study.

Below are detailed comments from our reviewers. These points are not merely suggestions but are critical for the scientific validity and integrity of your research. It is essential that each of these points is addressed thoroughly in your revision for further consideration of your manuscript for publication.

*Reviewer #1 (Recommendations for the authors):*

Thank you for the revisions made to your manuscript. While appreciative of your efforts, it is critical to address several key issues that remain outstanding. These revisions are necessary to ensure the scientific rigour and validity of your paper:

1) The lack of cross-validation or the use of leave-one-out (LOO) cross-validation in cases where the test set is representative only of <5% of the data under the reason of not having enough data while having 21 control and 35 OA participants. This is important since the OA population is highly heterogeneous and is difficult to make generalised assessments and statements based on such outcomes as the results can be clearly overfitting. Thus, it is still strongly felt that the authors have to improve the cross-validation and in cases where they feel that the data is not enough for the model use a simpler model such as a linear regression or a support vector machine with a proper cross-validation split for such analysis. Furthermore, if having slightly more or less data is such an issue for the model then this would be a raise for a further concern in any case.

2) It is imperative to include a comparative analysis with simpler models like logistic regression and SVM. This is necessary to substantiate the choice of your complex models and to address potential overfitting issues.

3) The difference in exclusion criteria for a percentage of missing data and Pearson r coefficient threshold between the control and OA population raises a major concern for making invalid comparisons. This was not corrected by the authors stating that because the data for patients were less the criteria differed, which is not a sufficient reason.

4) Discrepancies between the data clustering for slow and fast walking were also ignored and attributed to "likely just how the PCA algorithm processes the data", which is not correct. What is even more concerning is that now in the manuscript the figure looks different and the colours for fast and slow in Figure 4 C were switched without this being commented. This is a clear contradiction because the authors did not perceive the data to be wrong and any need to make any amendments, and yet the figure colour coding is switched for the very last tile of Figure 4.

5) Although information regarding the treatment groups is available online in Section 8.2.6.6 of the R5069-OA-1849 clinical study protocol, the manuscript will still benefit from a small clarification in the text regarding this as it is quite difficult for general readers to go through clinical studies for a brief detail like these.

6) Suggestions on the inclusion of a discussion regarding utilizing inertial measurement units for such an assessment as a cost-effective alternative for insoles, especially since walking speed parameters were identified as major markers, was also ignored and not stated to be included in the text.

7) Elaborations on the linear model fitted to each point along the vGRF curve (visualization of knee arthropathy signatures from vGRF data, Supplementary Figure 2) in the Methods Section were also not stated to be included in the text.

*Reviewer #2 (Recommendations for the authors):*

The authors have gone through the comments and addressed only some of them. However, a good amount of comments (from the other reviewers as well) remain. Please address them accordingly. For the comments I have given in the previous round, please see below for things that still need clarification:

1. Regarding the comment about the presence of osteoarthritis and the components of the ground reaction forces that are required for this. Indeed the insoles can only measure vertical GRF, but I believe a statement detailing the importance of the other components for OA should be added in the manuscript as a potential limitation.

2. I understand the study did not account for familiarization, but I would say it might be an important influencer. Therefore, at least adding a statement on this limitation might make the work clearer.

3. Please specify in the manuscript that the assumptions for the statistical tests were indeed checked for, as mentioned in the response to the reviewers' document.

---

## [Author Response]

Essential revisions (for the authors):1) There is a need for improved cross-validation. Specifically, clarify the number of cross-validation folds and method for data splits (subject-wise or across the whole dataset) for the XGBoost model, reporting both the mean and standard deviation for the cross-validation split. Furthermore, for the CNN, leave-one-out cross-validation across 44 participants, results in only 2.3% of the data used for a test set, which can very likely result in overfitting, thus needs to be more stringently tested for this.

We thank the reviewers for this comment, and have amended the manuscript to include, where relevant, mean and SD values for cross validation folds (e.g., see lines 255–256). When constructing the ML models with the force plate derived vGRF data, we used 5-fold cross validation in the training/validation set. This full training/validation set was then used to construct a single model and applied to the test set. As a note, we filtered out subjects with knee injury on both joints, and there was only one vGRF observation per subject.

When not enough data were available to perform cross validation, in the case of the digital insole derived data, XGBoost models were trained and assessed using leave-one-out cross-validation (LOOCV), where models are evaluated by iteratively leaving one subject out, building a model, and evaluating where that subject would be classified compared to the true result.

For your reference, Author response table 1 provides an overview of the models that were used in this study:

**Author response table 1. sa2table1:** 

Model	Train/Validation	Test	Number of CV folds	Figure
Left foot XGBoost vGRF model	N=471 total (85%)N=297 left foot injuryN=174 no injuryauROC = 0.905 (0.033)auPR = 0.938 (0.026)	N=84 total (15%)N=47 left foot injuryN=37 no injuryauROC = 0.857auPR = 0.895	5 folds	Figure 2d
		Digital insole vGRF (left)N=21 controlN=35 OAauROC=0.815auPR=0.862	N/A	Figure 2d/e
	Digital insole vGRF (left)N=21 controlN=35 OAauROC = 0.967auPR = 0.979	N/A	leave-one-out cross-validation (LOOCV)	Figure 4d/e
Right foot XGBoost vGRF model	N=427 total (85%)N=248 left foot injuryN=179 no injuryauROC = 0.929 (0.025)auPR = 0.954 (0.020)	N=76 total (15%)N=44 left foot injuryN=32 no injuryauROC = 0.863auPR = 0.920	5 folds	Figure 2—figure supplement 3
		Digital insole vGRF (right)N=21 controlN=35 OAauROC=0.859auPR=0.909	N/A	
	Digital insole vGRF (right)N=21 controlN=35 OAauROC = 0.951auPR = 0.952	N/A	leave-one-out cross-validation (LOOCV)	
Derived gait characteristics XGBoost	N=243 walks(N=21 subjects) controlN=73 walks(N=37 subjects)OAauROC = 0.9997auPR = 0.9989	N/A	leave-one-out cross-validation (LOOCV)	Figure 4d/e
Derived gait characteristics without highly correlated (rho>0.7) walking speed characteristics XGBoost	N=243 walks(N=21 subjects) controlN=73 walks(N=37 subjects)OAauROC = 0.998auPR = 0.993	N/A	leave-one-out cross-validation (LOOCV)	Figure 4d/e
Time series XGBoost	N=21 controlN=32 OA (subjects with either Day 1 or Day 85)	N/A	leave-one-out cross-validation (LOOCV)	Figure 4d/e
Time series CNN	N=21 controlN=32 OA (subjects with either Day 1 or Day 85)	N/A	leave-one-out cross-validation (LOOCV)	Figure 4d/e
Time series XGBoost at each timepoint	N=21 controlN=27 OA (subjects with at least Day 1)	N/A	leave-one-out cross-validation (LOOCV)	Figure 4f
Sensor models for CNN individuality model	N=10 controlN=13 OA (subjects with at least Day 1)	N=11 controlN=14 OA (subjects with at least Day 1)	N/A	Figure 5
Sensor models for CNN consistency model	N=11 OA (subjects with both Day 1 and Day 85)	N=12 OA (subjects with both Day 1 and Day 85)	N/A	Figure 6

2) Explicitly state how much data were discarded due to missing signal from one or more sensors or to outliers and regularity of repeated patterns (address concerns that only regular patterns were used, how this can skew the data, and why it was not done for the force plate data). Furthermore, why was the data from the clinical trial processed differently (different exclusion criteria for percentage of missing data and Pearson r coefficient threshold)?

For force plate data, no data were discarded for Figure 2, where we included all subjects with knee injuries on a single joint. We limited the analysis of the force plate dataset for ML model building to those with left or right knee injury (excluding subjects who had knee injury on both joints). The rationale for this was to more closely match the clinical trial enrollment criteria (KL score ≥2, index joint).

For digital insole raw sensor time series data, for control participants, 16.0% of walks were discarded due to missing data and features or being too short; a further 3.5% of walks and 19.7% of strides were excluded from the analysis because of low correlation with walks and strides from the same participant. For OA clinical trial participants, 7.6% of walks were discarded due to missing data; a further 8.3% of walks and 17.0% of strides were excluded from the analysis because of low correlation with strides from the same participant. We note that this is a similar percentage of discarded strides from both the control and OA groups.

We added the following sentence in the main manuscript to address the limitations (see lines 514–517): “The study's analytical approach, which focused solely on regular walking patterns and excluded outliers and irregular patterns in the insole gait data, may limit the comprehensiveness of the findings; future research should aim to collect more diverse walking data per subject to include non-regular patterns.”

Regarding data processing, given limited clinical trial data, a lower threshold was used for OA clinical trial participants, as they had data for only one walk on each Day 1 and Day 85. In contrast, the control participants each had multiple walks (and thus more data per participant). Although different thresholds were used, both control and OA clinical trial patients exhibited significant gait individuality compared to those in same disease class, suggesting that the range of thresholds chosen would not affect the conclusions.

However, we acknowledge that ideally identical thresholds would be used for all gait processing from the same device.

3) If possible, quantitatively compare the simultaneous recordings of the two systems (force plates and pressure insoles) or amend the statements regarding such comparison to reflect the concern and state that it is only qualitative.

This comment is addressed below. There were no simultaneous recordings of the two systems performed in this study.

4) State whether a correction for multiple comparisons was applied to the t-tests, and if it is not included, it should be applied and the new results should be reported.

For the Control vs OA comparisons (Supplementary File Table 1) we have added a column indicating FDR corrected p values (adjusted for multiple comparisons). In general, we have added language throughout indicating nominal vs FDR adjusted p values (e.g., see line 387).

Nominal p values are still quite informative in this context, even if they do not meet experiment-wide significance. The spirit of this work is largely hypothesis-generating, rather than specific hypothesis-testing (https://www.huber.embl.de/msmb/06-chap.html), and as such we feel strongly about using p values as one of several tools to understand the data properly. We are generating and screening digital biomarkers, and like other biomarkers where we make e.g., 10,000–100,000 of tests in a small number of subjects (less than N=50) we caution the reader not to over interpret any single test.

5) There is a need for an additional data analysis regarding the specification of the severity or type of gait impairment (can you classify different severity and discriminate between affected joints). This is vital when there are claims about disease signatures, or disease classifications, as otherwise, the results can reflect general cautious, slow, or antalgic gait rather than the disease.

Our R5069-OA-1849 clinical trial was not designed to discern associations between derived gait characteristics, captured by the Moticon digital insole, and imaging (K-L score) or Performance Outcome Assessments (PROs) like WOMAC metrics. As such, we chose not to present this analysis in the main paper since this may lead to over-interpretation of the results—we limited our conclusions to simpler disease vs healthy control identification, and the face validity of speed.

The closest we could come to doing this was an exploration of the interrelation between laterally derived gait characteristics (e.g., derived separately for the left or right insoles) and two laterally collected clinical measures—the WOMAC pain sub-score and the K-L imaging disease severity score. The relationships were independently evaluated for each joint. Other scores (e.g., the WOMAC physical function score) were not collected for both joints or were not collected at all.

We correlated each derived gait characteristic from the respective joint (either left or right) individually with the corresponding joint’s (A) K-L score, or (B) WOMAC pain sub-score (see Author response image 1). Although none of these correlations achieved experiment-wise significance, some of the traits demonstrated nominal significance for at least one joint, and investigation of the distribution of nominal spearman rho values between joints allows for further understanding of these relationships when viewed as a scatter plot—to assess if the relationship is similar between joints. Our reasoning was that if these were similar for each joint, they may represent a real trend.

**Author response image 1. sa2fig1:** 

This analysis has major limitations and has several assumptions and constraints, which is why we chose not to focus on this in our study. Of all the subjects in this study, only N=14 had knee-only OA. Thus, the study's findings could be confounded by the presence of other forms of OA (e.g., hip OA), which many of the rest of the subjects had. This also makes the (likely incorrect) assumption that joints act independently. A proper study to address this question would take into account: (1) asking subjects what their dominant foot is (e.g., left vs right footed), and (2) would control for pain on only a single joint (e.g., a KL score of 0 on at least one joint). We do not have this dataset.The suggestions for further analysis concerning the categorization of gait impairment severity or type are valuable. We agree that such analysis can enhance the specificity and relevancy of disease signatures and classifications, minimizing the risk of reflecting general cautious, slow, or antalgic gait rather than the disease itself. However, our current study was not designed with this in mind, but it indeed provides promising avenues for future research.

We have added the following sentence to the study limitations (see lines 510–511): “Further, the OA study did not control for knee-only OA on one or both joints.”

6) There is a need for further evidence against the possibility that the only thing that the model classifies is slow vs normal pace walking. For example, can a simple model trained on walking speed compare in performance to the current results?

Our study devoted significant attention to walking speed and its relationship with the derived gait characteristics. We aimed to explore the associations between derived gait characteristics and varying walking speeds in both control subjects and those with OA, not least because gait speed is highly clinically relevant, and different walking speeds have high face validity.

Our findings demonstrated that a cluster of parameters, including 14 derived gait characteristics, consistently correlated with walking speed using a conservative cutoff (|r|>0.7). Moreover, using principal component analysis (PCA) dimensionality reduction, we found a continuum linking the arthropathy state of knee OA and walking speed. It was evident that subjects with knee OA generally exhibited a slower walking pace.

However, we did not restrict our analysis to only classifying 'slow' versus 'normal' walking. In fact, we investigated a broader range of gait characteristics and aimed to ascertain if these features could enhance the accuracy of classifying knee OA relative to control, beyond the factor of speed alone. When we employed derived gait characteristics that excluded the 14 speed-correlated attributes, we achieved higher classification accuracy (auROC = 0.998, auPR = 0.993) than speed alone. To us, this suggests that a deeper understanding of gait can be captured with digital insoles, going beyond mere speed, to detect disease-specific features.

In response to your suggestion of comparing our model with a simpler model trained solely on walking speed, we have done this too. We found that walking speed alone could distinguish between knee OA subjects and healthy controls with substantial accuracy (auROC = 0.981, auPR = 0.983). While this result is promising, our objective was not just to establish a relationship between walking speed and disease status, but also to unearth additional gait characteristics that could enhance the discriminative power of our model.

We hope this provides further clarity and are open to further suggestions.

7) Compare the deep learning models developed here to the performance of simpler models like linear regression and support vector machine.

The choice of a model in machine learning is often somewhat arbitrary and is usually guided by the specific problem at hand, data characteristics, and the performance requirements for the scientific questions. In our study, we chose XGBoost for our machine learning model and a 1D Convolutional Neural Network (CNN) for our deep learning model.

The XGBoost model was chosen over other models like Support Vector Machine (SVM) and Random Forest (RF) for several reasons. XGBoost, being a gradient boosting framework, offers several advantages in terms of model performance and efficiency. Gradient boosting algorithms, including XGBoost, often outperform other algorithms, especially on datasets where the relationship between the features and the target variable is complex or involves non-linear relationships, as is the case in gait analysis. In the case of this dataset, the analysis was robust to the model choice and similar performances was observed with logistic regression and SVM approaches.

As for our deep learning model, we used a 1D Convolutional Neural Network (CNN). CNNs are particularly suited for our use-case because they excel in handling sequential data with temporal dependencies, such as time series data from our digital insoles. CNNs automatically learn and extract important features, reducing the need for manual feature engineering, and they are robust against shifts and distortions in the data, which makes them ideal for our application.

We acknowledge that comparing the performance of our chosen models with simpler ones like linear regression could offer additional insights. However, given the complexity and high-dimensional nature of our data, and the non-linear relationships it likely embodies, we believe that the chosen models are better suited to our task.

8) Provide further information and manuscript changes in accordance with the reviewers' comments.

Please see responses below.

Reviewer #1 (Recommendations for the authors):(1) The made equivalency of ground reaction forces (GRFs) obtained via force platforms (load cells or force transducers) and pressure insoles (pressure sensors) can be misleading (e.g. line 415). The same applies to comparing the abilities of the pressure insoles to those of force platforms by using one (vertical GRF – vGRF) of the multiple variables (3-axial forces and moments as well as centre of pressure) to compare their performance, while in some cases (raw sensor data and derived gait characteristics) utilizing all of the data from the pressure insoles. Although it can be understandable to use vGRF to compare to the one from pressure insoles (as this is the most consistent variable obtained with pressure insoles), it can lead the readers to believe that the system equivalence is apt. The statements equating measurements from these two systems and stating the outcomes of such analysis should be amended to reflect that.

This is an interesting point. While GRF may not be calculated or obtained in the exact same manner between force plates and insole, force plates are the accepted reference device (in the field) for any pressure-based gait outcomes from wearable devices, and therefore we feel are appropriate for comparisons.

2) Similarly, the comparison of vGRF between force platforms and pressure insoles is only qualitative (demonstrating that the two measurements look similar in the control case) and even then, the pattern for knee OA is significantly different from the one of knee arthropathies demonstrated from the force platforms. If possible, can simultaneous recordings of the two systems (force plates and pressure insoles) be quantitively compared, or if such data does not exist in the dataset, amend the statements regarding the comparison to reflect that?

This is a great suggestion; however, our current experimental design and dataset do not allow for us to perform this analysis. Although we did not conduct the study ourselves, Moticon Insoles have been quantitatively compared to force plates, with the aim to verify the analytical validity of the insoles' force output in the context of walking at normal speed. This type of analytical validation has been performed by Moticon, and is publicly available here: https://moticon.com/wp-content/uploads/2021/05/Moticon_SCIENCE_total_force_validation.pdf

Quantitative comparison has been done in other research and the comparison of force plates and insoles was not an objective of this study. The study involved multiple walking sequences of N=10 subjects, captured using two Hawkin Dynamics force plates, used with Moticon insoles, under various conditions designed to reflect everyday use. Each walking sequence consisted of ten rounds repeated three times. The measurement protocol was designed to accommodate the "warming up" phase of the insoles, and steps were taken to limit external variables, such as using the same model of shoe for all participants.

3) Digital insole data pre-processing:a. In the Methods Section, it is stated that since the digital insoles do not collect data in regular intervals the data was interpolated. This is confusing since data collection intervals should not matter as the analysis is not done in real-time. Can you clarify what is meant by this? Does it mean that the data is sampled at 100 Hz but these 100 samples per second are not regular (i.e. the first 100ms contain 20 samples while the next 100ms contain 5 samples) or is it meant that the data was up-sampled to 1000 Hz from 100 Hz to match the force plate data sampling rate? Also, state the force plate sampling rate and the brand/model of the equipment.

Thank you for your comment. To clarify, the Moticon insoles do sample at a consistent rate of 100 Hz. However, the number of samples per specific output can vary based on the walking speed of the individual. For instance, a faster walking speed may result in fewer samples per second for a specific walking curve, while a slower speed may result in more samples per second. The data was interpolated to account for these variations and ensure a consistent number of data points for analysis. The force plate data sampling rate is much higher, and this information is available as part of the GAITRec dataset.

b. Further in the section, it states that walks with missing data from 1 or more sensors, or with missing data from 1% of the walk were discarded. Can you state how much data was discarded as a percentage of the overall data?

For control participants, there was an initial total of 773 walks across all participants and walk speeds. Two (0.3%) walks were discarded because of missing features, 115 (14.9%) walks were discarded because of more than 1% missing data, and seven (0.9%) walks were discarded because they were less than 10 strides.

For OA clinical trial participants, there was an initial total of 65 walks across all participants. Five (7.6%) walks were excluded because of more than 5% missing data.

c. Similarly, data was discarded not only based on outliers but also on the regularity of repeated patterns. How much of the data was discarded as a percentage of the overall data in this part? Also, this is skewing the data to consider only regular patterns, which should be explicitly stated when comparing results and discussing them as it is a vital detail and was also not a thing done for the force plate data.

For control participants, 649 walks and 10,512 strides across all participants and walk speeds remained after removing those with missing data or having too few strides in a walk. Of these, 23 (3.5%) walks and 2,067 (19.7%) strides were discarded because of low correlation with other strides and walks from the same participant.

For OA clinical trial participants, 60 walks and 6,929 strides across all participants remained after removing those with missing data. Of these, 5 (8.3%) walks and 1,177 (17.0%) were discarded because of low correlation with other strides and walks from the same participant.

We do say in the Methods section (Digital insole raw sensor time series data processing) the following:

“*Segments that had that an outlier number of samples (outliers defined as 1.5*iqr +/- q3 or q1) were excluded, such that only regularly repeating segments, or strides, were analyzed*.” See lines 784–785.

“*Under the assumption that an individual’s strides within a walk should be highly regular to each other, each stride’s Pearson r correlation with the means of the remaining strides was computed (stats.pearsonr), and any strides with an outlier Pearson r correlation (outliers defined as 1.5*iqr +/- q3 or q1) were excluded*.” See lines 789–782.

In addition, the following sentence has been added to the Limitations section of the manuscript (see lines 514–517): “*The study's analytical approach, which focused solely on regular walking patterns and excluded outliers and irregular patterns in the insole gait data, may limit the comprehensiveness of the findings; future research should aim to collect more diverse walking data per subject to include non-regular patterns.*”

d. Why was the data from the clinical trial processed differently (different exclusion criteria for percentage of missing data and Pearson r coefficient threshold)? These have to be the same no matter what for comparative reasons.

Given limited clinical trial data, a lower threshold was used for OA clinical trial participants, as they had data for only one walk on each Day 1 and Day 85. In contrast, the control participants each had multiple walks. Although different thresholds were used, both control and OA clinical trial patients exhibited significant gait individuality compared to those in the same disease class, suggesting that the range of thresholds chosen would not affect the study conclusions.

e. Can you clarify more on the treatment population groups and the number of participants? Line 587 states 13 patients per treatment group aiming for a total of 30 patients, but the different treatment groups were not stated.

As stated in Section 8.2.6.6 of the R5069-OA-1849 clinical study protocol (available here: https://classic.clinicaltrials.gov/ct2/show/NCT03956550), eligible participants for the study were men and women ≥40 years of age with a clinical diagnosis of OA of the knee based on the American College of Rheumatology criteria with radiologic evidence of OA (K-L score ≥2) at the index knee joint as well as pain score of ≥4 in WOMAC pain sub-scale score. There were a total of three treatment groups in the study, which included a standard dose of the investigational product (IP) REGN5069 administered intravenously every 4 weeks, a high dose of REGN5069 administered in the same way, and a matching placebo given intravenously every 4 weeks, with patients randomized into each group at a 1:1:1 ratio. Approximately N=13 patients per each treatment group were targeted for enrollment in the sub-study evaluating Moticon insoles, with the aim of collecting data from at least N=10 patients per group (around N=30 patients for the entire sub-study). Since REGN5069 lacked clinical efficacy (there was no statistically significant change in the WOMAC score or any of the subcomponents relative to placebo), the analysis for this manuscript did not focus on the treatment effects of the IP.

The primary study for this sub-study has now been published, please see Somersan-Karakaya Selin, et al. Monoclonal antibodies against GFRα3 are efficacious against evoked hyperalgesic and allodynic responses in mouse join pain models but, one of these, REGN5069, was not effective against pain in a randomized, placebo-controlled clinical trial in patients with osteoarthritis pain. (*Neurobiology of Pain*. 2023;14:100136. https://doi.org/10.1016/j.ynpai.2023.100136.)

4) Can you comment on the pressure insoles size and fit for the participants, were there any cables connecting the insole leads to devices on other parts of the body, were the fit and the presence of insoles affecting the normal gait of the participants?

In our pilot study, we did find that overall, the Moticon insole was user friendly and could be comfortably worn for short periods of time. Out of the 22 participants who were involved in the pilot study, 17 participants (77%) reported that it was comfortable to wear the Moticon insole to walk. Among the other five participants, three reported that they could feel the battery in the foot arch, and two reported that they did not typically wear insoles and the insole made their shoes tight and hot. All participants commented that they did not see issues wearing the insole for a short period of time. Please note that this survey was not quantitative nor intended to make general assumptions—the major strength of the insoles in our view is their differentiation from force plates, and this work shows that even when worn for short periods of time, clinically useful information can be obtained from them.

5) Concerning the derived gait parameters – It is explicitly mentioned in the Methods Section that the investigated gait parameters were less influenced by walking speed, but the Results section states that walking speed correlates were excluded rather than that they were significantly reduced. It is an important distinction, especially since the threshold was "Spearman rho > 0.7".

This is an excellent point, and we thank the reviewer for the feedback related to walking speed.

With respect to the 14 derived gait characteristics that strongly correlate to walking speed (|Spearman rho|>0.7), we have modified the text when they are first introduced to read (see lines 283–288): “Correlations within and between categories of parameters revealed that similar groups of parameters clustered together (Figure 3b), including 14 derived gait characteristics strongly correlating with walking speed (|Spearman rho|>0.7). We note that many other gait characteristics will be influenced by walking speed, and as such, we call out these as representing the subset most influenced by speed, defined by this threshold.”

Walking speed's influence on gait characteristics are well documented. Our mention of the correlation threshold serves to delineate the subset of gait characteristics that are markedly influenced by walking speed, aiding in the focused analysis of gait parameters less impacted by speed variations.

Similar studies to ours have employed Principal Component Analysis (PCA) to reduce data and segregate gait characteristics into distinct domains based on their strongest correlations, not dissimilar to our approach. As such, we currently feel comfortable with the edits to delineate speed (or pace) from other gait characteristics.

Related references:

Morris R, Hickey A, Del Din S, Godfrey A, Lord S, Rochester L. A model of free-living gait: A factor analysis in Parkinson’s disease. *Gait and Posture.* 2017;52:68–71. ISSN 0966-6362.

Stuart S, Parrington L, Morris R, Martini DN, Fino PC, King LA. Gait measurement in chronic mild traumatic brain injury: A model approach. *Hum Mov Sci*. 2020;69:102557. doi: 10.1016/j.humov.2019.102557. Epub 2019 Nov 26. PMID: 31783306.

Verghese J, Wang C, Lipton RB, Holtzer R, Xue X. Quantitative gait dysfunction and risk of cognitive decline and dementia. *J Neurol Neurosurg Psychiatry*. 2007;78(9):929–35. doi: 10.1136/jnnp.2006.106914. Epub 2007 Jan 19. PMID: 17237140; PMCID: PMC1995159.

Lord S, Galna B, Verghese J, Coleman S, Burn D, Rochester L. Independent domains of gait in older adults and associated motor and nonmotor attributes: validation of a factor analysis approach. *J Gerontol A Biol Sci Med Sci*. 2013;68(7):820–7. doi: 10.1093/gerona/gls255. Epub 2012 Dec 18. PMID: 23250001.

6) Lack of model cross-validation folds information:a. For the XGBoost model – the force plate dataset was randomly split into 85% training and 15% testing datasets. Which force plate dataset was used (GaitRec?) and, more importantly, how many times was this repeated (how many cross-validation folds)? Was the split done across participants (85% of participants for training and 15% for testing) or across the whole dataset? If it is for the whole dataset, can you report a participant split? This will indicate how general the model is and testify to its strength to be used in a typical scenario when you want to predict OA in new patients.The mean and standard deviation (SD) for the cross-validation needs to be reported. Furthermore, this trained model was further tested using the R5069-OA-1849 clinical study dataset only, is that correct?

We have amended the manuscript to include, where relevant, mean and SD values for cross validation folds (e.g., see lines 255–256). The only force plate dataset that we had available to use was the GaitRec dataset. For the force plate vGRF dataset, it was first randomly split with 85% samples to be used for training/validation and the remaining 15% put aside as a hold-out test set. Five-fold cross validation was used to initially assess the model performance on the 85% training/validation set. This full training/validation set was then used to construct a single model and applied to both the 15% hold-out test set and the digital insole-derived vGRF data from the R5069-OA-1849 clinical sub-study. As a note, there was only one vGRF observation per subject in the vGRF datasets (both from force plates and from digital insoles), so there were no repeated participants between the folds/splits.

b. Similarly, leave-one-out cross-validation (LOOCV) was applied for the convolutional neural network (CNN) model but there are no mentions of how many folds. How many participants, in total, was this model trained on and did you cycle over all of them (number of folds = number of participants) or was this done only once? If done only once, it needs to be repeated for all participants and the results for the mean and SD from these need to be reported.

To clarify, the LOOCV was indeed applied for the CNN model, and the number of folds equaled the number of participants minus one. Each participant was used as a left-out test participant in one model, resulting in N different CNN models for N participants. Each model was trained on the other N-1 participants. This process was repeated for all participants, not just once. The results reported are the mean and standard deviation from these iterations (described in detail in the Methods section, see lines 863 on).

c. Furthermore, the CNN results need to be tested for overfitting. For example, if the LOOCV of the CNN included 44 participants, then 1 participant represents only ~2.3 % of the data, meaning that the model most likely overfits this dataset. Thus, this needs to be checked by including more participants in the cross-validation (i.e. 85%/15% split across participants). This is raised since the CNN performance for the classification of knee OA using derived gait characteristics achieves auROC = 1 and auPR = 0.999, which can be strongly indicative of overfitting and needs to be checked.

Thank you for this comment, and we appreciate the deep review of this modeling approach. We acknowledge your concern about potential overfitting given the high performance of our CNN model. However, we believe that our approach using LOOCV, where each participant was used as a left-out test participant in one model, is a robust method to evaluate the model's performance. This method ensures that the model is tested on unseen data, reducing the likelihood of overfitting.

While we agree that testing the model with a larger split, such as an 85%/15% split across participants, could provide additional insights, we believe that our current approach and sample size are sufficient to answer our research questions. The high auROC and auPR values, while potentially indicative of overfitting, could also suggest that our model is highly effective at classifying knee OA using derived gait characteristics.

d. Why was the cross-validation split different for the individuality and consistency CNN models and why is the last output consisting of a 23-element vector (shouldn't that be 44 to reflect the 44 different participants from the R5069-OA-1849 clinical study?)?

For individuality and consistency CNN models, where the penultimate layer was extracted as a gait signature, the size of the last layer was set to the largest number of individuals in the training set, which was 23 (10 control and 13 OA. See Figure 5).

7) The results for subject-specific gait signatures are very exciting. However, the further claims on the ability to identify clinical attributes beyond knee disease as well as generalizability of the method are not substantiated. The results demonstrate that if data is included from both days of recording (day 1 and day 85 of treatment), there is a trend (P=0.033) suggesting it could be possible to identify the subject from others even across days. However, for this to work, you need training data from both timepoints of treatment, which is not the overall aimed application. It is still great that the distinction between individual participants is preserved regardless of the treatment time point! Although, the statement for the generalization across days has to be amended to reflect the results as the results from training on day 1 only reflect that it cannot distinguish between the day 85 data from the same participant and the rest. Similarly, the claim for a richness of data to help identify other clinical attributes is not corroborated by these results since they indicate that one is able to find different classification boundaries between participants. This does not necessarily reflect differences in clinical attributes as they could be solely based on different walking patterns and symmetry in pressure distribution.

Thank you for the enthusiasm regarding this finding—we share the excitement. Our understanding of this comment is that it relates to the claim that “clinical” attributes not yet examined may be in the data, in which case we agree that the statement is confusing. We have amended the statement to read as such (see lines 846–849): “If we could train models to identify individual subjects from their walk, or from just a single stride, this may suggest that the gait data collected has at a minimum that ability to identify attributes beyond knee disease.”

We also note that models could be built in a variety of ways (e.g., training on both Day 1 and Day 85, or more generally two timepoints), to answer a variety of questions. We hope that providing this data publicly will encourage others to investigate such questions (with this data or their own).

8) Statistics – was there a correction for multiple comparisons for the t-tests? If so, that should be stated in the methods and, if not, it should be included and reported. Can you also overlay statistical significance on the plots?

Please see above, this comment has already been addressed.

9) Can you include comments on the additional benefit of using pressure insoles instead of inertial measurement units (IMUs) and include classification results using just the acceleration and angular velocity data? In the manuscript, it is mentioned a few times that gait speed (a feature that can be derived from a much cheaper IMU) is one of the main features to help classify OA. The further analysis in the paper demonstrates that features that are less correlated with speed can achieve a better performance, which is great. However, it will be good to quantify explicitly the IMU role and clarify further what else could be possible by having the additional pressure data, which can excite the reader even more and give further perspective.

We appreciate the reviewer's insightful suggestion. We acknowledge the potential of IMUs as a cost-efficient alternative to force plates in certain applications. However, clinically, our focus is on gait including weight bearing in an OA population. Changes in gait patterns result in a shift in weight transfer/bearing in OA/replacement, which the insoles can accurately capture, unlike IMUs. The additional information provided by insoles includes the ability to determine when the foot is actually on the ground via raw pressure sensor data, as opposed to IMUs that require algorithms to estimate the initial and final contact of the foot, from which gait metrics (e.g., gait speed / velocity) are calculated (Paraschiv-Ionescu A, Soltani A, Aminian K. Real-world speed estimation using single trunk IMU: methodological challenges for impaired gait patterns. *Annu Int Cone IEEE Eng Med Biol Soc.* 2020;2020:4596–4599).

We concur with the reviewer that supplementing vGFR with IMU data can provide a more comprehensive characterization of the kinematics and dynamics of walking. This could potentially enhance our understanding of the disease mechanisms.

10) Similarly, can you elaborate more on the importance of identifying participants using the CNN approach investigated in this work? There is a sentence about this on line 405 but it would be beneficial for the reader to understand more and will show the importance of your results.

Gait data is extremely rich in substance, and the focus of this work is on demonstrating the utility of the gait data we collected to address specific questions. We can speculate but want to focus primarily on the results we have and on the general utility of thinking about how data from studies are used with different types of models to address various questions.

11) The inclusion of the foot as an important dependable for classification in the XGBoost model (line 269) should be stated to be because the OA participants were injured only unilaterally (left leg), thus, this will generate these differences regardless and should be put as a confound. Have you tried classifying only from the "healthy" leg in OA participants, to demonstrate that the affected side does not matter?

Note, there are no “healthy” legs in the OA dataset. This would also (incorrectly) assume that the legs act independently. As stated in the manuscript and above in this response, OA disease is highly heterogeneous making this type of analysis difficult. Figure 4f is presented to be exploratory.

12) Can you comment on the fact that knee OA participants are walking more slowly, which is indicated by the clustered data for vGRF and less by the gait characteristics, but the raw sensor data clusters the OA right next to the fast walking control group (Figure 4 A-C)? What causes this different and counterintuitive clustering?

This observation is likely just how the PCA algorithm processes the data (which are related but do contain different types of information).

13) Can you elaborate on the linear model fitted to each point along the vGRF curve (visualization of knee arthropathy signatures from vGRF data, Supplementary Figure 2) in the Methods Section?

A major question that came up in this work related to how vGRF are influenced by other obviously important covariates. Given that the vGRF curves are normalized in such a way that allows for their comparison across individuals (and this analysis was performed on a very large sample size from the force plat dataset), we wanted an analysis that captured whether there were other covariates with potentially stronger signal than arthropathy state. The use of linear models at each of the 98 points during the precent stance phase allowed us to examine the relationship between vGRF and the covariates (disease, age, sex, and body weight) at each specific point in time during the stance phase of walking. This is important as the relationship between these variables and vGRF may change throughout the stance phase.

We do feel that our conclusion, that arthropathy state is likely a more important factor than the three other covariates tested, is supported by this analysis. However, we also feel that there may be other analytical approaches to more appropriately address this question—ours is simple and straightforward, and hopefully intuitive to any statistician.

Reviewer #3 (Recommendations for the authors):– Insoles have been used for gait assessment in multiple groups, and validation studies exist. The introduction might benefit from a summary of these findings.

We have added references to this point in the introduction.

– In the introduction, a rationale could be provided for why knee arthropathy was chosen for the current paper. In the eyes of the authors, what does this group reflect? Please also discuss previous studies regarding GRF in knee OA (e.g. Costello et al., Osteoarthr Cartilage 2021; Costello et al., Br J Sports Med 2023).

Our decision to focus on knee arthropathy was a decision related to our drug development program, primarily one hypothesis that pain relief could potentially alter gait, thereby increasing disease progression. The current paper represents the initial part of a series of experiments designed to explore this hypothesis. We have updated the Introduction significantly to reflect the focus on knee arthropathy. However, we note that this is a secondary digital biomarker analysis, and as such we chose this initially for exploratory efforts with wearable measurements.

– In the methods, I missed information about which knee injuries were included. In line with my previous comment, did they all have pain above a certain threshold? This group can be very heterogeneous.

This is a good point and has been addressed in the manuscript and in response to comments above. To emphasize, this is a highly heterogeneous population, with a diverse set of K-L and WOMAC scores on both the index joint, as well as other joints.

– The set-up of having only one group with disease vs controls, is severely limiting the potential impact of the ML model. If the authors would be able to test if this model could be trained to differentiate between pain levels, or which joints are involved, that would substantially improve the impact. See for example Leporace et al. Clin Biomech 2021.; Kushioka et al. Sensors 2022.

We agree that expanding the model to differentiate between pain levels or involved joints could indeed enhance its impact (that being said, we feel that our analysis sufficiently addresses the questions that we pose, and further, we feel like these are important questions). We will certainly consider these aspects in our future research. Thank you for pointing us towards the studies by Leporace et al. and Kushioka et al., they will be valuable references for us. While we acknowledge the limitations of our current dataset, we believe our analysis provides meaningful insights into the questions we initially set out to answer. We look forward to further refining our model and exploring these additional avenues in our subsequent work.

[Editors’ note: what follows is the authors’ response to the second round of review.]

After detailed discussions with our review panel, it is clear that significant concerns remain unaddressed. These issues are critical for the integrity and publication viability of your work.Key areas requiring your immediate attention include:1. Methodological Rigour: There are serious concerns about your choice of cross-validation method and modelling approach, given the heterogeneity of the OA population. A comparative analysis with simpler models is also lacking, raising questions about overfitting.

Additional work related to methodological rigour has been incorporated into the main text and the methods to address these questions. Most notably, we benchmarked different model choices and added in additional types of cross-validation to address concerns about potential overfitting. These updates are further described below in Reviewer #1; Responses 1-2.

2. Data Presentation and Analysis: Discrepancies in data handling and presentation, particularly in Figure 4 and your exclusion criteria, are worrying. These issues cast doubts on the validity of your comparisons and findings.

A figure version control issue has been addressed, described below in Response 4.

3. Comprehensive Explanation: Essential details are missing or inadequately explained. This includes the need for clearer elaboration on the linear model in the Methods Section and a more thorough discussion on the limitations and assumptions of your study.

Details that were previously only in the response to reviewers or not addressed have now been added directly into the main text of the document. Additionally, a more comprehensive study limitations section has been written.

Below are detailed comments from our reviewers. These points are not merely suggestions but are critical for the scientific validity and integrity of your research. It is essential that each of these points is addressed thoroughly in your revision for further consideration of your manuscript for publication.Reviewer #1 (Recommendations for the authors):Thank you for the revisions made to your manuscript. While appreciative of your efforts, it is critical to address several key issues that remain outstanding. These revisions are necessary to ensure the scientific rigour and validity of your paper:1) The lack of cross-validation or the use of leave-one-out (LOO) cross-validation in cases where the test set is representative only of <5% of the data under the reason of not having enough data while having 21 control and 35 OA participants. This is important since the OA population is highly heterogeneous and is difficult to make generalised assessments and statements based on such outcomes as the results can be clearly overfitting. Thus, it is still strongly felt that the authors have to improve the cross-validation and in cases where they feel that the data is not enough for the model use a simpler model such as a linear regression or a support vector machine with a proper cross-validation split for such analysis. Furthermore, if having slightly more or less data is such an issue for the model then this would be a raise for a further concern in any case.

We thank the reviewer for raising this point. To assess this, we compared the results of leave-one-out cross-validation (LOOCV) to repeated (n=50) 5-fold cross validation (r5FCV). We found that the results were highly concordant between the two evaluation approaches, as detailed in Author response table 2.

**Author response table 2. sa2table2:** 

	auROC	auPR		
	LOOCV	r5FCV	LOOCV	r5FCV
vGRF	0.984	0.988	0.990	0.992
Derived gait characteristics	0.997	0.996	0.988	0.986

These additional statistics have been included in the main text on page 13.

We also hope that, to the best of our ability, this addresses questions of generalizability from a modeling perspective. The OA population is indeed heterogeneous, and further studies (i.e., more sample size) will be the best way to address this, but we feel we have made every attempt to not overstate conclusions with respect to the generalizability, as well as to incorporate this into the study limitations.

2) It is imperative to include a comparative analysis with simpler models like logistic regression and SVM. This is necessary to substantiate the choice of your complex models and to address potential overfitting issues.

We thank the reviewer for this suggestion. While all of these three methods (Logistic Regresion, SVM, XGBoost) had comparable performance in the model training, the SVM approach appeared to be the most generalizable with improved test set performance. As a result, we have switched the main results reported in the text to using a SVM model. Additionally, we have added in the model benchmarking to the methods on page 38 and included the figures summarizing these results below, which have also added as a supplemental figure (figure 3 – supplement 4).

3) The difference in exclusion criteria for a percentage of missing data and Pearson r coefficient threshold between the control and OA population raises a major concern for making invalid comparisons. This was not corrected by the authors stating that because the data for patients were less the criteria differed, which is not a sufficient reason.

Thank you for your feedback about these thresholds. We have redone the data-processing and analysis using the same thresholds for both populations. Specifically, we modified the data-processing thresholds for the control population to match the OA population, such that for both populations a walk was excluded if it had more than 5% missing data or if the mean of Pearson r coefficients fell below 0.9. We have updated the methods text describing these thresholds.

For control participants, there was an initial total of 773 walks across all participants and walk speeds. 2 (0.3%) walks were discarded because of missing features, 7 (0.9%) walks were discarded because less than 10 strides, and 6 (0.8%) walks were discarded because of more than 5% missing data. Previously, 115 (14.9%) walks were discarded because of more than 1% missing data, so loosening the missing data threshold resulted in keeping 109 walks at this stage. 758 walks and 12,250 strides across all participants and walk speeds remained after removing those with missing data or having too few strides in a walk. Of these, 15 (2.0%) walks and 2,330 (19.0%) strides were discarded because of mean correlation less than 0.9 with other strides and walks from the same participant. Previously, 23 (3.5%) walks and 2,067 (19.7%) strides were discarded because of mean correlation less than 0.95 with other strides and walks from the same participant. At the end of this processing, 743 walks and 12,296 remained, compared to 626 walks and 10,558 before, so loosening both thresholds resulted in keeping an additional 117 walks and 1,738 strides.

The data processing of walks from OA population remained as before. Analyses and figures (Figures 4 and 5) that rely on the raw data processing of the control walks have been updated accordingly. These changes did not affect our conclusions.

4) Discrepancies between the data clustering for slow and fast walking were also ignored and attributed to "likely just how the PCA algorithm processes the data", which is not correct. What is even more concerning is that now in the manuscript the figure looks different and the colours for fast and slow in Figure 4 C were switched without this being commented. This is a clear contradiction because the authors did not perceive the data to be wrong and any need to make any amendments, and yet the figure colour coding is switched for the very last tile of Figure 4.

We apologize for this oversight which was accidentally introduced due to figure version controls. In the last round of review, the version of Figure 4 that was embedded into the text showed the correct colors, while the uploaded version of the figure mistakenly had the colors inconsistent with the legend in panel C, which occurred due to the figures being generated by two different analysts and with a common legend shared across panels A-C.

The correct version of the figure is now uploaded. Notably, the slow walking speed was nearest to the OA group across each of the different panels A-C. ­

5) Although information regarding the treatment groups is available online in Section 8.2.6.6 of the R5069-OA-1849 clinical study protocol, the manuscript will still benefit from a small clarification in the text regarding this as it is quite difficult for general readers to go through clinical studies for a brief detail like these.

Thank you for this comment, and we are of course happy to provide this information. The Study Design in the Methods section was amended to read as follows:

“This information is available publicly in the protocol in Section 8.2.6.6., Moticon Digital Insole Device Sub-Study for Gait Assessments. All patients in this sub-study were enrolled at two study sites in the USA and Moldova and the study was conducted between June 2019 and October 2020. The date of first enrollment in the R5069-OA-1849 trial 17 June 2019, and last patient visit was 29 October 2020. The sub-study targeted to enroll approximately 13 patients per treatment group to obtain data on at least 10 patients per treatment group for a total of approximately 30 patients across the entire sub-study. The treatment groups were as follows: patients were randomized in a 1:1:1 ratio to receive a low dose of REGN5069 at 100 mg IV every 4 weeks (Q4W), and a high dose of REGN5069 at 1000 mg IV Q4W, or matching placebo Q4W.”

6) Suggestions on the inclusion of a discussion regarding utilizing inertial measurement units for such an assessment as a cost-effective alternative for insoles, especially since walking speed parameters were identified as major markers, was also ignored and not stated to be included in the text.

We apologize for this oversight. We have added this to the section of the discussion prior to the study limitations. We concur with the reviewer that supplementing vGFR with IMU data can provide a more comprehensive characterization of the kinematics and dynamics of walking. This could potentially enhance our understanding of the disease mechanisms, where the specific disease and application matter. The following has been added to the discussion to capture these tradeoffs on lines XX to XX:

“A strength of using digital insoles over other lower cost technology (e.g. IMUs) is that they can accurately capture the gait cycle and weight transfer/bearing in OA patients, which was our primary clinical focus here. Unlike IMUs that require algorithms to estimate the initial and final contact of the foot, digital insoles provide additional information such as the ability to determine when the foot is actually on the ground via raw pressure sensor data, thereby improving the accuracy of gait event detection and proceeding outcomes. This is the reason why digital insoles are often used for analytical validation of IMU algorithms in clinical populations. Although digital insoles may not be as hardwearing as IMUs (i.e., capacitors may deteriorate over time, so are not recommended for free-living assessment), they are useful clinical tools for active mobility tasks in an OA population, as they provide gait and pressure related metrics that are relevant for this clinical condition. Alternatively, in free living settings (e.g., passive monitoring with wearables over hours / days / weeks), digital insoles may have issues with device placement and sensor deterioration, unlike IMUs that have been shown to be comfortable and have high acceptance across various clinical populations.”

7) Elaborations on the linear model fitted to each point along the vGRF curve (visualization of knee arthropathy signatures from vGRF data, Supplementary Figure 2) in the Methods Section were also not stated to be included in the text.

We apologize for this oversight and have now added a section to the methods describing the linear model and elaborating on this (text below). We hope that this will give enough detail for the general statistically interested reader to understand our approach, and its limitations.

“Linear Models to associate covaraites with vGRF signal

Using the GaitRec force plate dataset, consecutive linear models were fit at each of the 98 % stance phase timepoints. We used disease (knee arthropathy or control), age, sex (male or female), and body weight as covariates in the model (lm and anova function, R), with each subsequent vGRF % stance phase timepoint as the dependent variable. Within each linear model, using the sum of squares for each category divided by the total sum of squares, we calculated the variance of each component’s contribution to the total variance, with the residuals indicating the unexplained variance in these models. The use of linear models at each of the 98 points during the % stance phase allowed us to examine the relationship between vGRF and the covariates (disease, age, sex, and body weight) at each specific point in time during the stance phase of walking. This is important as the relationship between these variables and vGRF may change throughout the stance phase.”

Reviewer #2 (Recommendations for the authors):The authors have gone through the comments and addressed only some of them. However, a good amount of comments (from the other reviewers as well) remain. Please address them accordingly. For the comments I have given in the previous round, please see below for things that still need clarification:1. Regarding the comment about the presence of osteoarthritis and the components of the ground reaction forces that are required for this. Indeed the insoles can only measure vertical GRF, but I believe a statement detailing the importance of the other components for OA should be added in the manuscript as a potential limitation.

We have added some detail to the limitations section of the manuscript regarding medial and lateral GRF relevance in OA, and a future direction of study for insole technology development:

“Additionally, the insoles are currently only capable of providing vGRF outcome measures, but medial and lateral GRF would also be useful outcomes for OA, as these also relate to pain and OA severity. Technology developments would be required to develop a wearable system that would be capable of capturing comprehensive GRFs during walking in OA, and should be an area of future research.”

Citation: https://www.oarsijournal.com/article/S1063-4584(21)00664-6/fulltext

2. I understand the study did not account for familiarization, but I would say it might be an important influencer. Therefore, at least adding a statement on this limitation might make the work clearer.

We agree, and have added the following sentence to the study limitations:

“The study design also did not specifically account for familiarization with the insoles.”

3. Please specify in the manuscript that the assumptions for the statistical tests were indeed checked for, as mentioned in the response to the reviewers' document.

We have added this in the Methods:

“Throughout, assumptions of the t-test were checked through creation of univariate histograms of each variable to qualitatively test for normality (i.e., does the data look Gaussian).”